

# Evaluating Unsaturated Hydraulic Conductivity Models for Diverse Soils and Climates: A Functional Comparison of Additive, Junction, and Kosugi Parameterizations

Asha Nambiar[1], Gerrit Huibert de Rooij[1]

Helmholtz-Centre for Environmental Research - UFZ, Halle (Saale), 06120, Germany

*Correspondence to*: Gerrit Huibert de Rooij (gerrit.derooij@ufz.de)

**Abstract.** Soil water moves as capillary flow, film flow, and vapour diffusion. The additive model for the unsaturated soil hydraulic conductivity curve adds up conductivities of capillary water, adsorbed water, and an equivalent vapour conductivity.

A recently introduced junction model with liquid water in either films or capillaries has one parameter less. We compared calculated water fluxes based on both models and Kosugi's model (which only considers capillary water) by fitting the RIA soil water retention curve and the three conductivity models to data for three soils. Five subsets of the model parameters were calibrated by fitting, with the other parameters fixed. For all 135 resulting cases, we ran the Hydrus-1D numerical model for uniform columns of these soils subjected to generated weather records for three climates. Hydrus-1D crashed 14 times for the

additive model, twice for Kosugi's model and once for the junction model. The conductivity models and fitting parameter sets only significantly affected the drainage flux at 2 m depth (and, in one case, the transpiration), but the effect was only large in two cases. If the conductivity models disagreed, the additivity model was usually the outlier. An analysis of the water balance terms revealed the impact of soil type to be limited on transpiration or infiltration, but stronger on groundwater recharge. The conductivity models had only a minor effect. The fluxes were insensitive to differences in the dry-range conductivity. Fewer

fitted parameters rarely altered the results significantly. This favours the more parsimonious and robust junction and Kosugi models.

## 1 Introduction

The presence of liquid water in soils is governed by the soil water retention curve (SWRC) (Hillel, 1990, p. 142–161), and its movement by the unsaturated hydraulic conductivity curve (UHCC) (Hillel, p. 203–233). Recently, a new

parameterization for the SWRC was developed (de Rooij et al., 2021; de Rooij, 2022, 2024a) with a distinct air-entry value, a sigmoidal intermediate range, and a logarithmic dry range ending at zero water content for a finite matric potential. The logarithmic dry branch (modelled after Rossi and Nimmo, 1994) removed the asymptotic residual water content that appears for instance in parameterizations of Brooks and Corey (1964) and van Genuchten (1980). Fuentes et al. (1991) showed that constraints on some of the parameter values were needed to avoid that asymptotic functions could lead to the storage of an





infinite amount of water in a finite soil column, which is obviously non-physical. These constraints were such that they would not be met by many soils. For van Genuchten (1980), the requirement was that shape parameter $n > 2$, for instance.

The distinct air-entry value (adopted from Ippisch et al., 2006) removed the risk of non-converging integrals in Mualem's (1976) model for the UHCC for which Durner (1994) warned. If that occurs, the slope of the UHCC becomes infinite when the matric potential $h$ (L) is zero. Ippisch et al. (2006) showed this is the case if $n < 2$ in van Genuchten's (1980)

parameterization, making it impossible to have a value for $n$ in which both the dry end and the wet end of the curve behave in a physically acceptable way. Madi et al. (2018) generalized the criterion developed by Ippisch et al. (2006) to guarantee a finite slope of the UHCC at saturation, applied it to 18 parameterizations of the SWRC, and found that only four of them satisfied the requirement.

De Rooij et al. (2021) tested the new SWRC parameterization and compared its performance to those of van

Genuchten (1980) and Ippisch et al. (2006). They found that simulations with Hydrus-1D (Šimůnek et al., 2013, 2016; PC-progress website) of soil water flow in different one-dimensional soil columns under different climates converged more often and faster, and gave more plausible results than some produced by the curves of Ippisch et al. (2006).

Many models for the UHCC have been proposed over the years. Assouline and Or (2013) provided an insightful review. Several of the reviewed UHCC models can be cast in the general from of Eq. (8) of Kosugi (1999), which is a model

with a single scale parameter (the soil hydraulic conductivity at saturation) and three shape parameters. The models of Burdine (1973), Mualem (1976), and Alexander and Skaggs (1986) all arise as special cases by setting the shape parameters to specific values. The model of Assouline (2001) requires two shape parameters to be zero, leaving one scaling and one fitting parameter free.

The UHCC models discussed above treat all water as if it resided in water-filled capillaries. Tuller and Or (2001,

2002) introduced a model with different pore geometries aligned in parallel in which liquid water could completely fill capillary tubes with a triangular cross-section, or retract to the corners of such tubes (behind a curved air–water interface) and in a thin film on the tube walls between the corners (behind a flat air–water interface). Parallel to the tubes, there were slits separating two flat solid surfaces. Here too, water could fill the slits completely, or retract into thin films behind a flat air–water interface. Flow velocities could be calculated for full tubes and slits, for the corners in partially filled tubes, and for the water films in

partially filled tubes and slits. The increased viscosity in thin films was accounted for. The hydraulic conductivity for any combination of tube and slit sizes could be found from the various contributing flows. Or and Tuller (2000) used a comparable approach to model the hydraulic conductivity of unsaturated fractures with uneven surfaces.

This parallel arrangement of different configurations was adopted in a different form by Peters and Durner (2008), Peters (2013), and Weber et al. (2019). They dropped the distinction between capillary-bound water in saturated tubular pores,

saturated slits, and corners of unsaturated tubular pores. Film flow conductivity was a power law of the degree of saturation (Peters and Durner, 2008) or a power law of the scaled matric potential (Peters, 2013), with the exponent of the power law adopted (Peters, 2013) or adapted (Weber et al., 2019) from Tokunaga (2009). Peters (2013) and Weber et al. (2019) included water vapour diffusion, and expressed it as an equivalent hydraulic conductivity. The constituting conductivities were added,



in the case of Durner and Peters (2008) and Peters (2013) after weighting. De Rooij (2024b) pointed out that adding domain

conductivities relies on the assumption of a parallel arrangement of these domains, and derived UHCCs based on different types of averaging to represent different domain configurations. In doing so, he also demonstrated that it is fundamentally impossible to identify procedures for averaging or adding domain conductivities that can correctly represent the configuration of the domains in the soil. De Rooij (2024c, 2025) therefore introduced a junction model for the UHCC in conjunction with the junction model of the SWRCC of de Rooij (2022, 2024a). In the dry range, all water is assumed to occur as water films,

and in the wet range, all water moves through saturated capillaries. This simpler parameterization required one parameter less than the more complicated models. It may also avoid the numerical problems with evaluating the equation for capillary conductivity identified by Heinen (2023).

De Rooij (2025) found that an unweighted additive model and the junction model were sufficient to provide good fits for most of the 13 soils on which he tested the junction model and those of de Rooij (2024b). There was evidence that the

seven parameters of the unweighted additive model (de Rooij, 2024b) led to overparameterization, although, for a few soils, all parameters were needed to achieve good fits. For those soils where these models performed poorly, the UHCC models based on different averaging methods (de Rooij, 2024b) did not fare better.

De Rooij's (2025) test only considered the quality of the fit of the UHCC model to measured conductivity values. The main purpose of parametric models of the SWRC and UHCC is to represent soils in numerical solvers of Richards'

equation for unsaturated flow that are used to address scientific and practical problems involving movement of water and solutes in soils, often related to crop growth and groundwater recharge (Šimůnek and Bradford, 2008; Heinen et al, 2024). A functional test, in which the different models are used by a numerical model to calculate water fluxes in different soils under different conditions, is still lacking. The objective of this paper is to provide such a test for Kosugi's classical UHCC model (KGV), the unweighted additive model (ADV), and the junction model (JUV), all of them including diffusive movement of

water vapour. The test consists of an evaluation of the fluxes and multi-year soil water balances generated by a numerical model, and on the ability of the numerical model to complete model runs without crashing. The test uses the same set of problems that de Rooij et al. (2021) devised for their test of three SWRC models, and will also use the Hydrus-1D numerical model to solve Richards' equation.

## 2 Materials and Methods

The functional test was set up as a set of model simulations that each cover several years, similar to that used by de Rooij et al. (2021). We refer to that paper and its supplement for details of the simulations and the weather records that were used for the upper boundary condition, and only summarize the main characteristics here.

Weather data (daily rainfall and potential evapotranspiration rates) were generated for a monsoon, a temperate, and a semi-arid climate. Uniform columns of 2.00 m height for three different soils were vegetated with grass (10 cm height). As de

Rooij et al. (2021) noted, this is not realistic for the semi-arid climate but it facilitates the comparison between soils and



climates. The root zone was 50 cm deep, with a uniform root distribution (Brown et al., 2010). Interception was neglected. Surface ponding was allowed up to a depth of 1.0 cm. Heat flow was not considered. Vapour flow was included through the inclusion of a vapour component in the UHCCs of the three soils, not by using the vapour flow feature in the numerical model that was used for the study. The artificial weather data provide the upper boundary condition. We selected the Atmospheric

BC with surface runoff from the Hydrus-1D options. During rainfall, the boundary condition switches from a prescribed flux to a prescribed head ($h = 0$) as soon as the precipitation rate exceeds the infiltration rate. At the lower boundary, unit-gradient flow was imposed. The initial condition was hydrostatic equilibrium with $h$ equal to zero at the bottom of the soil column. The simulated period lasted 16 years, of which the final 10 were used to avoid effects of the initial condition.

### 2.1 Soils

We selected three soils from those used by de Rooij (2025) that had good fits to both the soil water retention and the hydraulic conductivity data, and had contrasting properties (Weber et al., 2019): Pachepsky's soil (sandy loam), SM–35–119 (silt), and UNSODA 2571 (loamy sand), with the texture classification according to USDA (Gee and Or, 2002). Figures 1–4 show the SWRC of de Rooij (2022) and the UHCC according to three models fitted by de Rooij (2025) for the three soils. Table 1 gives the parameter values of the fits of the SWRC used in the simulations. In this table and elsewhere, $\theta_s$ denotes the

saturated water content, $h_{ae}$ and $h_d$ are the fitted matric potentials (L) at air-entry and oven-dryness, respectively, and $\alpha$ (L$^{-1}$) and $n$ are shape parameters. For the UHCC, five different combinations of fixed and fitted parameters were fitted for each model. These are (de Rooij, 2025):

    1. all six or seven fitting parameters $h_{ae}$, $\alpha$, $n$, $K_{s,c}$, $K_{s,a}$ (for ADV), $\gamma$, and $\tau$ fitted

    2. $h_{ae}$, $h_j$, and $h_d$ as for the SWRC; $\alpha$, $K_{s,c}$, $K_{s,a}$ (for ADV), $\gamma$, and $\tau$ fitted

3. $\tau = 0.0$ (Assouline, 2001); $h_{ae}$, $\alpha$, $n$, $K_{s,c}$, $K_{s,a}$ (for ADV), and $\gamma$ fitted

    4. $\gamma = 2.0$, $\tau = 0.5$ (Mualem, 1976); $h_{ae}$, $\alpha$, $n$, $K_{s,c}$, $K_{s,a}$ (for ADV) fitted

    5. $h_{ae}$, $\alpha$, and $n$ as for the SWRC; $K_{s,c}$, $K_{s,a}$ (for ADV), $\gamma$, and $\tau$ fitted

| Table 1. Parameter values of the fitted soil water retention curves for three soils. | | | |
|---|---|---|---|
| Parameter | Pachepsky | SM–35–119 | UNSODA 2571 |
| $\theta_s$ | 0.42888 | 0.44402 | 0.48077 |
| $h_{ae}$ (cm H$_2$O) | −2.0202 | −8.1778 | −4.0782 |
| $h_d$ (cm H$_2$O) | −6.3094 · 10$^6$ | −6.3032 · 10$^6$ | −6.3094 · 10$^6$ |
| $\alpha$ (cm$^{-1}$) | 1.3754 · 10$^{-2}$ | 4.2922 · 10$^{-2}$ | 0.36943 |
| $n$ | 1.4797 | 1.1881 | 1.3199 |
| $c$ | 3.0518 · 10$^{-5}$ | 1.0071 · 10$^{-3}$ | 3.0518 · 10$^{-5}$ |





Tables 2–4 only give the fits that had the lowest value of Akaike's information criterion, corrected for small sample sizes (AICc; Hurvich and Tsai, 1989). These were used in the model simulations. Above, in Tables 2–4, and elsewhere, $K_{s,c}$ and $K_{s,a}$ (LT$^{-1}$) are the hydraulic conductivities at saturation for the capillary and film domain, respectively (de Rooij, 2024a, 2025), and $\gamma$ and $\tau$ are exponents of different terms in the expressions for the UHCC (de Rooij, 2024a, 2025). Note that $h_d$ needs to be multiplied by a correction factor equal to $1+c$ to find the value at which the water content truly reaches zero (de

Rooij, 2022). Table 1 also lists $c$.

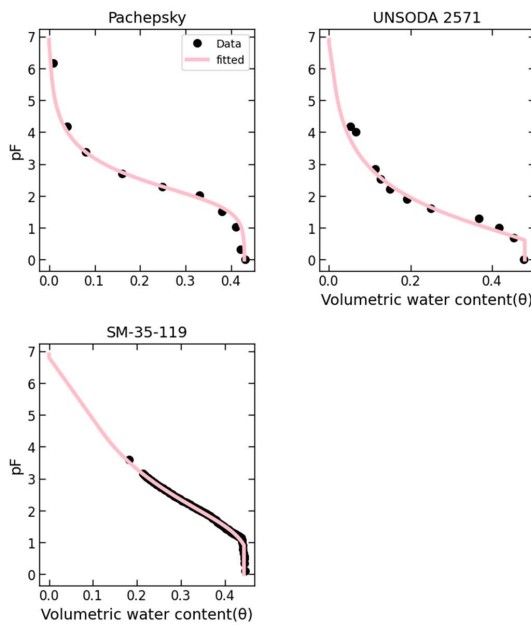

**Figure 1: Observed and fitted SWRCs of the soil used in the evaluation of the models for the UHCC curve. Abbreviations are listed in Appendix B.**




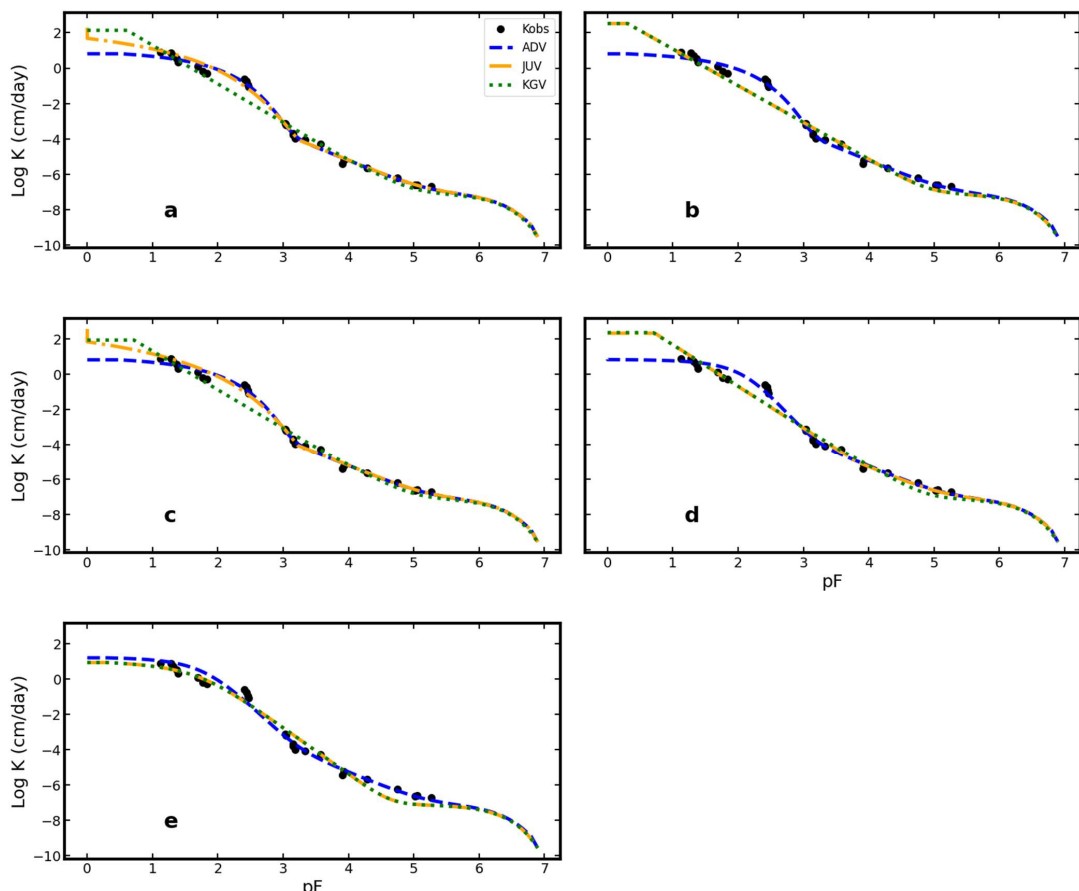


**Figure 2: Observed and fitted conductivity data for the three hydraulic property function models applied to Pachepsky's soil for five different combinations of fixed and fitted parameters, according to the numbered fitting parameter sets in Section 2.1 (a): set nr. 1, (b): set nr. 2, (c): set nr. 3, (d): set nr. 4, (e): set nr. 5.**



Table 2. Parameter values of three models for the unsaturated soil hydraulic conductivity curve for Pachepsky's soil (sandy loam). ADV: unweighted additivity model; JUV: junction model; KGV: Kosugi's (1999) model. All models include vapour flow. The number of the fitted parameter set (see Section 2.1) is given in the column headings.

| Parameter | ADV (set 2) | JUV (set 3) | KGV (set 4) |
|---|---|---|---|
| $h_{ae}$ (cm H$_2$O) | $-2.0202$ | $-8.6033 \cdot 10^{-3}$ | $-4.9977$ |
| $\alpha$ (cm$^{-1}$) | $7.6701 \cdot 10^{-4}$ | $6.6793 \cdot 10^{-4}$ | $2.5559$ |
| $n$ | $1.4797$ | $1.1214$ | $1.1425$ |
| $K_{s,c}$ (cm d$^{-1}$) | $6.4348$ | $3.5420 \cdot 10^{2}$ | $2.2321 \cdot 10^{2}$ |
| $K_{s,a}$ (cm d$^{-1}$) | $2.1376 \cdot 10^{-3}$ | $-$ | $-$ |
| $\gamma$ | $6.9046$ | $6.1964$ | $2.00000$ |
| $\tau$ | $0.13559$ | $0.00000$ | $0.50000$ |



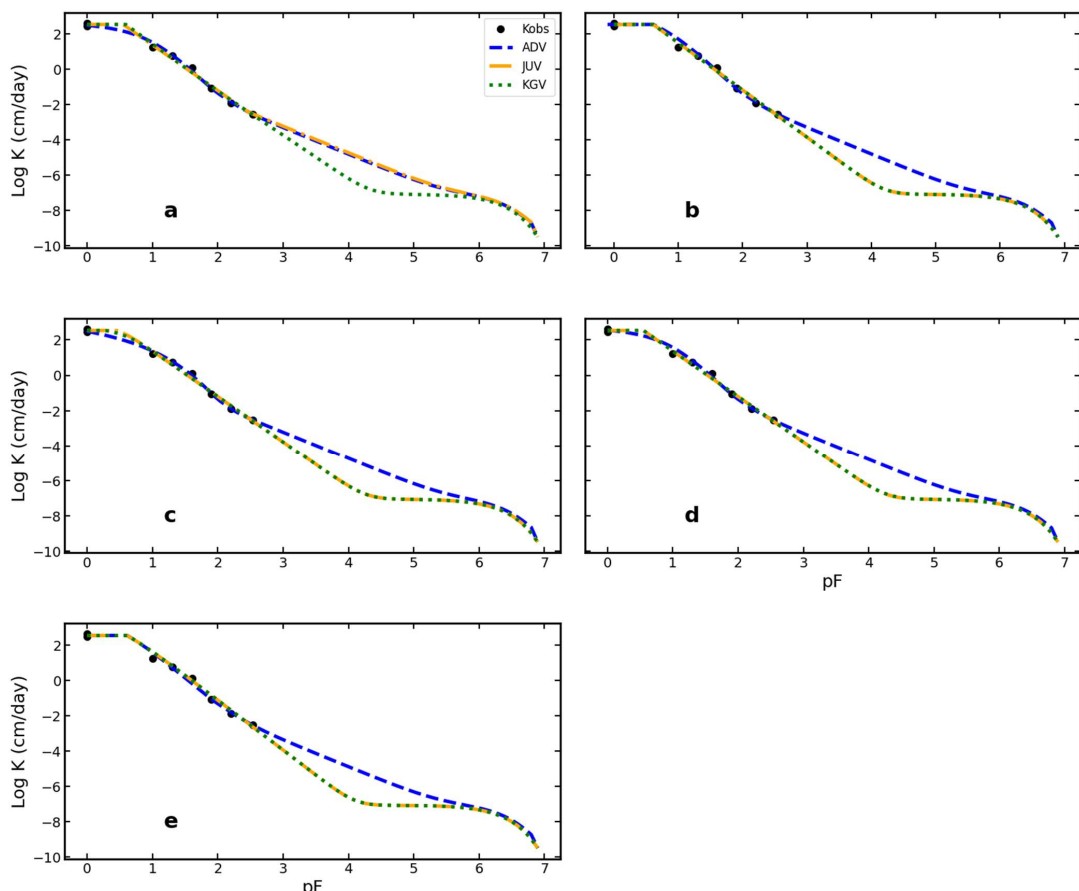

**Figure 3: As Fig. 2, for the UNSODA 2571 soil.**





| Parameter | ADV (set 5) | JUV (set 5) | KGV (set 5) |
|---|---|---|---|
| $h_{ae}$ (cm H$_2$O) | − 4.0782 | − 4.0782 | −4.0782 |
| $\alpha$ (cm$^{-1}$) | 0.36943 | 0.36943 | 0.36943 |
| $n$ | 1.3199 | 1.3199 | 1.3199 |
| $K_{s,c}$ (cm d$^{-1}$) | $3.4378 \cdot 10^2$ | $3.4076 \cdot 10^2$ | $3.4074 \cdot 10^2$ |
| $K_{s,a}$ (cm d$^{-1}$) | $8.1106 \cdot 10^{-3}$ | − | − |
| $\gamma$ | 1.0432 | 2.1028 | 2.1027 |
| $\tau$ | 5.4542 | 0.16667 | 0.16667 |

Table 3. As Table 2, for soil UNSODA 2571 (loamy sand).





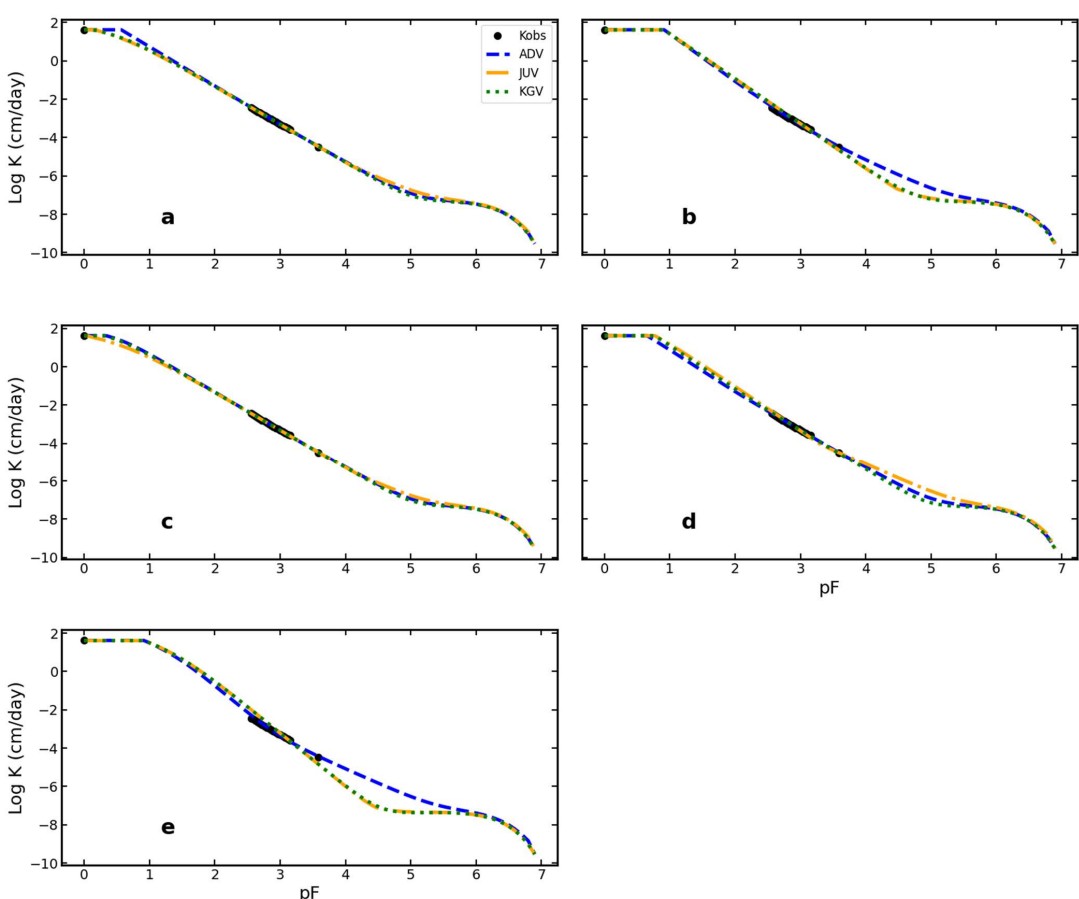

**Figure 4: As Fig. 2, for the SM–35–119 soil.**





| Table 4.  As Table 2, for soil SM–35–119 (silt). | | | |
|---|---|---|---|
| Parameter | ADV (set 4) | JUV (set 3) | KGV (set 3) |
| $h_{ae}$ (cm H$_2$O) | −4.4804 | −1.0793 | −2.0343 |
| $\alpha$ (cm$^{-1}$) | 1.2998 | 0.19597 | 0.33777 |
| $n$ | 1.0960 | 1.1658 | 1.8435 |
| $K_{s,c}$ (cm d$^{-1}$) | 40.882 | 40.878 | 40.878 |
| $K_{s,a}$ (cm d$^{-1}$) | $5.9861 \cdot 10^{-4}$ | – | – |
| $\gamma$ | 2.00000 | 1.6960 | 1.0721 |
| $\tau$ | 0.50000 | 0.00000 | 0.00000 |

## 2.2 Weather scenarios

145       The weather of the monsoon climate was based on weather data from Colombo (Sri Lanka), that of the temperate climate on data from Ukkel (Belgium), and that of the semi-arid climate on data from Tamale (Ghana). The weather statistics in Table 5, which summarizes a more detailed table of de Rooij et al. (2021), are based on generated weather records of 1000 years. Figure 5 gives the daily rainfall records used in the simulations.



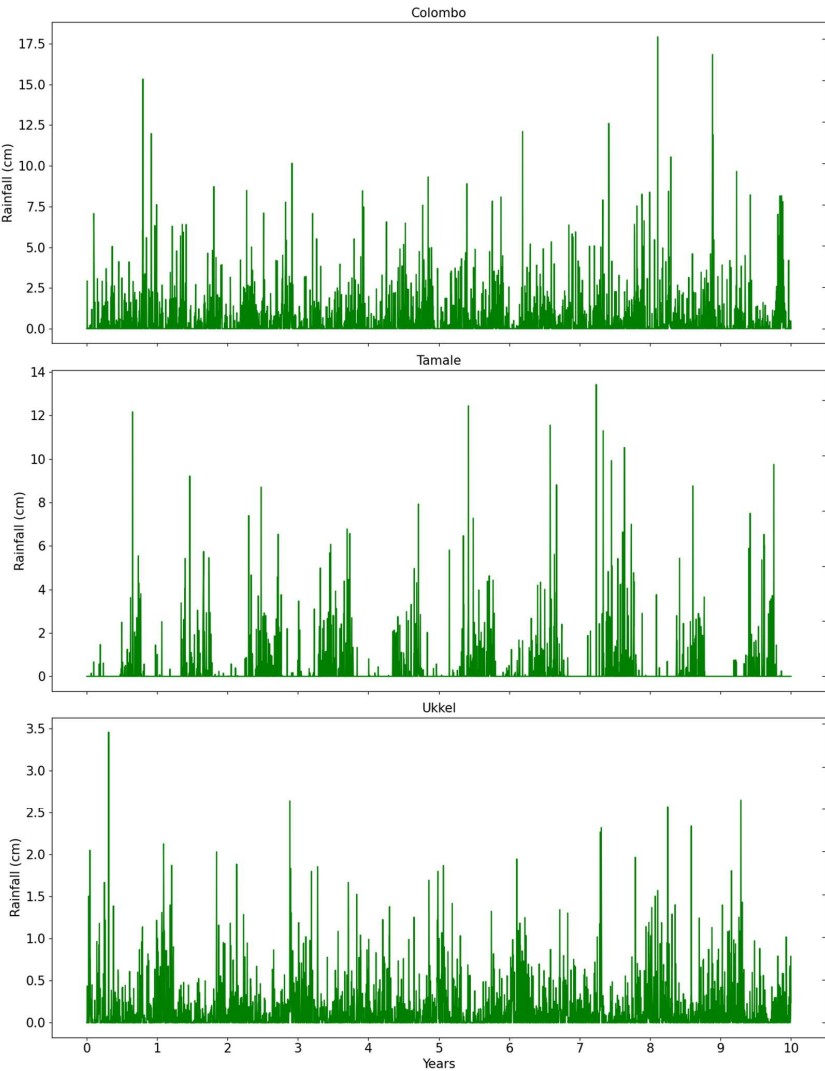

**Figure 5: Daily rainfall data for monsoon, semi-arid and temperate climate for ten-year period starting at January 1st.**



Table 5. Average cumulative 3-month and annual rainfall and potential evapotranspiration ($ET_{pot}$) calculated from 1000-year generated weather records for three climates.

| Period | Monsoon | | Semi-arid | | Temperate | |
|---|---|---|---|---|---|---|
| | Rain (mm) | $ET_{pot}$ (mm) | Rain (mm) | $ET_{pot}$ (mm) | Rain (mm) | $ET_{pot}$ (mm) |
| Jan–Mar | 394.9 | 483.3 | 40.9 | 502.5 | 184.0 | 65.9 |
| Apr–Jun | 609.7 | 488.7 | 292.6 | 495.7 | 106.0 | 208.3 |
| Jul–Sep | 409.5 | 511.5 | 570.0 | 435.1 | 67.9 | 210.6 |
| Oct–Dec | 766.6 | 421.3 | 102.4 | 455.3 | 138.3 | 58.1 |
| Annual sum | 2180.5 | 1904.8 | 1005.8 | 1888.7 | 496.1 | 542.9 |


### 2.3 Simulations with the numerical model

The one-dimensional simulations for all combinations of three soils and three climates were carried out by model Hydrus-1D, version 4.17.0140 (Šimůnek et al., 2013; PC-progress website). The output data from the KRIAfitter code was used as an input for a Mater.in file, which functions as a lookup table for soil hydraulic properties in Hydrus-1D.

155        The simulated period was 16 years. The initial time step for the numerical solution was set to 0.001 days, with minimum and maximum time step values set to their default values. The soil was uniform over the simulation depth of 2.00 m, with a unit-gradient lower boundary condition. Heat flow and vapour flow were not considered. Daily variations in transpiration and sinusoidal variations in precipitation were generated by Hydrus-1D. The total number of nodes was 151. The nodal distance was smallest near the surface (4 mm), and gradually increased to 19 mm as the profile extended deeper, reaching

the region below the root zone. Within the lower 5 cm of the profile, the nodal distance decreased progressively, reaching 11 mm at the bottom. The initial condition prescribed the matric potential everywhere in the profile, while the atmospheric boundary condition and free drainage were used as the upper and lower boundary conditions, respectively. A maximum ponded layer of 1.0 cm was permitted before excess water on the soil surface was discharged.

Simulations were conducted for a total of nine combinations of the three soils and three climates described above.

With three conductivity functions that needed testing and five sets of fitted parameters for each of them, we carried out a total of 135 model runs. This elaborate set-up allowed us to separate the effects of the soil type, the climate, and the chosen UHCC model. The runs for five different fits helped us to see if the variation in model output caused by different combinations of parameter values for the same UHCC model fitted to the same data had a significant effect on calculated soil water fluxes.

### 2.3 Data Post-Processing

170        We processed the model output using the Python programming language within the Jupyter Notebook environment (Executable Books Community, 2020), which offers useful Python libraries for data manipulation, analysis, and visualization.



The Hydrus output provides mostly cumulative fluxes, which we transformed to non-cumulative form to gain better insight in the dynamics of the soil water flow. The resulting time series of fluxes were smoothed by the Savitzky–Golay filter. This filter reduced noise while preserving the essential trends in the data through polynomial fitting over a moving window without losing
resolution (Press et al., 1992, p. 644–649).

In order to examine the effect of the UHCC models on extreme fluxes, we aggregated the daily unfiltered fluxes across the soil surface (transpiration, evaporation, and infiltration) and the lower boundary (termed deep drainage below) in non-overlapping time windows of 1, 5, 10, and 30 days. We calculated the annual extremes of these fluxes and their SDs for every aggregation window.

The effect of the UHCC models on the overall water balance was quantified by taking the cumulative fluxes across the soil surface and the bottom boundary, as well as the storage change for the 10-year simulation period after the burn-in period. These terms of the soil water balance were then made dimensionless by scaling them by the total rainfall.

## 3 Results and discussion

### 3.1 Performance of Hydrus-1D for different UHCC models

Of the 135 model runs across three soils, three climates, three conductivity functions and five sets of fitting parameters, one (0.74 %) failed for JUV, two (1.48 %) for KGV, and fourteen (10.37 %) for ADV. Out of 45 simulations for Pachepsky's soil, one (2.22 %) failed for JUV, two (4.44 %) for KGV, and eleven (24.44 %) for ADV. In the case of SM–35–119, three (6.66 %) failed for ADV while JUV and KGV always ran to completion. For UNSODA 2571, all models ran
successfully. For Pachepsky's soil, ADV failed to converge for all climates with Mualem's parametrization (fitting parameter set nr. 4 in Section 2.1). When all soil water retention parameters were fixed (fitting parameter set nr. 5 in Section 2.1), only the temperate climate achieved successful runs for all three UHCC models for Pachepsky's soil, while all three UHCC models crashed for the semi-arid climate. The average simulation runtime was 21.1 s for JUV, 21.8 s for KGV, and 16.8 s for ADV. These times include the failed simulations, so that the frequent crashes of ADV brought down the average run time.

### 3.2 Fluxes at the soil surface, through the vegetation, and at the bottom of the profile

The graphs in Appendix A show the filtered fluxes for all model runs. The Savitzky–Golay filter (Press et al., 1992, p. 644–649) sometimes distorts the signal near boundaries or sharp transitions due to over-smoothing or edge effects. This causes unrealistic spikes in the time series that can be safely ignored. We summarize the main findings from the Appendix here.

The transpiration becomes increasingly less noisy as the climate becomes drier, probably reflecting the increasing difficulty with which roots can extract water from a drier soil, reducing the ability to respond swiftly to fluctuations in the potential evapotranspiration. The surface flux (infiltration or actual evaporation) is almost exclusively determined by the



weather, with negligible effects of the soil type, UHCC model, or fitting parameter set. In contrast, the bottom fluxes are clearly affected by the soil type. The effect of the choice of the UHCC model and the parameter set on the bottom flux is more pronounced than for the other fluxes, but still limited, except for Pachepsky's sandy loam in the semi-arid and temperate climates.

The simulation results for JUV and KGV are often strikingly similar, with those for ADV at times deviating from them. The main exception is the bottom flux in the semi-arid and temperate climates for Pachepsky's soil, for fitting parameter sets 1 (All fitted) or 3 (Assouline) (Figs. A8–9).

In all other cases (Figs. A17, 26), the similarity of the bottom fluxes for the semi-arid climate using different UHCC models is noteworthy, since this is the flux that is most strongly affected by the soil hydraulic properties. The simulations show that even when the soils are quite dry for part of the year, the dry branch of the UHCC does not appear to have much effect on the soil water flux.

The mostly limited effect of the choice of fitting parameter set suggest that it often will not be necessary to fit all model parameters in order to obtain valid results. This offers opportunities to reduce the risk of overfitting and non-uniqueness.

### 3.3 Extremes of mean annual flux values across climates and soils

Figure 6 shows the average values of annual maxima of simulated evaporation (positive values) and infiltration (negative values) with their respective SDs for the various aggregation windows. The absolute values of the means are much larger for the 1-day than for the 5-day windows, and so are the SDs. Thus, the annual maxima reveal considerable short-term (< 5 days) variations in both fluxes. Between the larger aggregation windows, the relative reductions in the mean maxima and their SDs are consistently much smaller, indicative of a lack of prolonged periods of consistently high evaporation or infiltration.

In comparison to these overall trends, the differences between the UHCC models are small, and when they occur, they do so predominantly in the 1-day window, and more so for the mean than for the SD (UNSODA 2571 for Colombo and Tamale). The effect of the soil on the transpiration maxima is small for the temperate climate (Ukkel), larger for the monsoon climate (Colombo) and largest for the semi-arid climate (Tamale). For infiltration, the mean maxima are largely soil-independent in Colombo and Tamale, but SDs vary for different soils. For Ukkel, there is a clear effect of the soil type for the 1-day window.

Overall, climate dominates the fluxes at the soil surface, with minimal soil influence. The presence of cracks could alter these results, but that is beyond the study's scope. ADV occasionally differs from the other models, but not significantly.

The similarity between JUV and KGV observed in the filtered fluxes (Section 3.2) is evident here as well, with the largest deviation occurring for the 1-day maximum transpiration for UNSODA 2571 in Tamale's semi-arid climate. The ADV model differs at times from the other two mostly in the 1- and 5-day windows for Colombo and Tamale. The differences with



the means of the other two UHCC models sometimes exceed 10 %. The KGV model deviates by that much from the other two
models once (UNSODA 2571, Tamale, 1-day window), JUV never.

Figure 7 shows the mean and SD of the annual maximum transpiration rate. In Ukkel, these are nearly independent
of soil type and UHCC model. In Colombo and Tamale, Pachepsky's soil has larger maxima (particularly in Tamale) with
more year-to-year variation compared to the other soils. The choice of UHCC model has a minimal effect on the annual

maximum transpiration statistics.

Finally, Fig. 8 focuses on the annual flux density across the lower boundary. It is expected that this water will
eventually replenish the groundwater. Because intra-annual variations are of little interest for aquifers with their characteristic
times of years or decades (de Rooij, 2013), we only look at the range of the annual fluxes over the 10-year simulation period.
The UHCC models show mostly similar results, with ADV differing from JUV and KGV only for UNSODA 2571 in all

climates.

In Colombo, the large precipitation surplus results in groundwater recharge ranging from 25 to 100 cm per year. The
soil type has a clear effect on both the minima and the maxima, with the ratio between the extremes remaining about the same
around 0.34. Pachepsky's soil shows the smallest maxima, while UNSODA 2571 has the largest.

In Tamale, recharge is less than half that of Colombo, with a smaller ratio (around 0.30) of maximum and minimum

indicating more variation between wet seasons (the only time of the year with meaningful fluxes at 2 m depth). The soil type
strongly influences recharge, with about a twofold difference between Pachepsky's soil and UNSODA 2571.

For Ukkel, maximum recharge is nearly same as Tamale, despite a wetter climate. The wetter climate results in higher
minimum recharge values, except for UNSODA 2571, reflected in a higher ratio (0.38 to 0.44) of minimum and maximum
groundwater recharge. The soil has a notable effect, though less so than in Tamale.






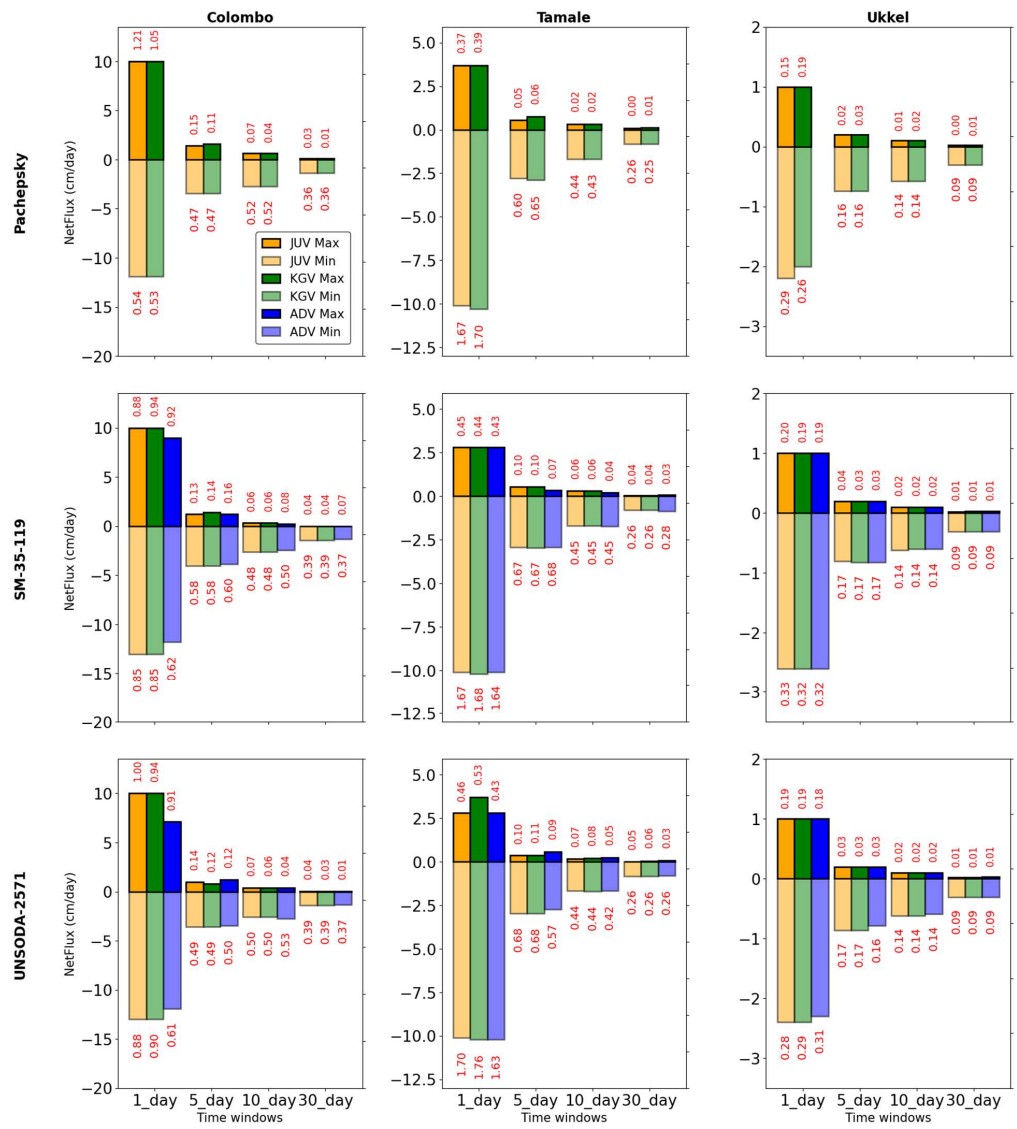


**Figure 6:** Average of annual maximum and minimum surface flux values aggregated over 1-day, 5-day, 10-day, and 30-day intervals, computed for three climatic conditions and three soil types during the simulation period. The SD values are displayed in red.





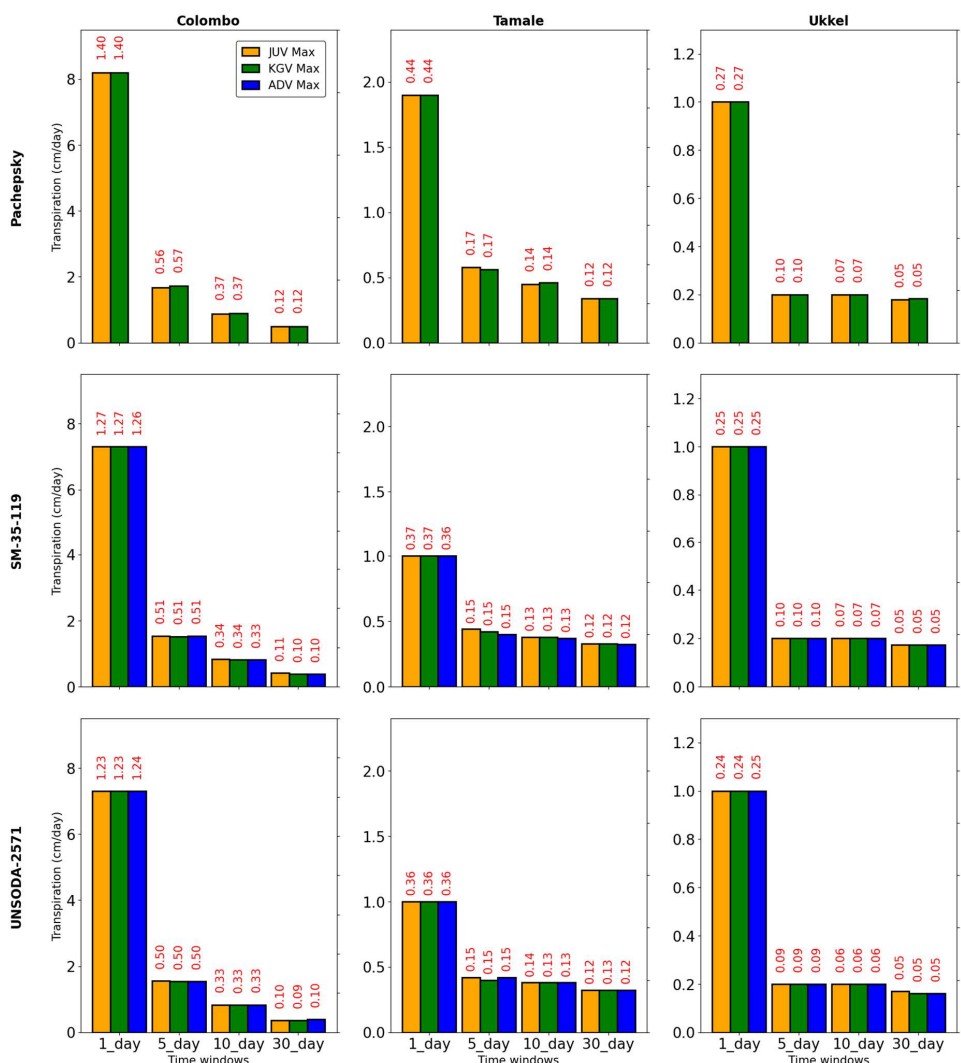

Figure 7: Average of annual maximum transpiration values aggregated over 1-day, 5-day, 10-day, and 30-day intervals, computed for three climatic conditions and three soil types during the simulation period. The SD values are displayed in red.







**Figure 8: Maximum and minimum mean annual bottom boundary flux values computed for three climatic conditions and three soil types during the simulation period. The ratio of maximum to minimum flux values are displayed in red.**




### 3.4 The soil water balance scaled by total rainfall

Within the terrestrial hydrological cycle, the soil partitions infiltrating rain water into various fluxes that carry water
back into the atmosphere (of which transpiration drives the vegetation), directly to surface water as overland flow, or to the
groundwater. Figure 9 shows the annual averages of the various fluxes over the 10-year simulation period, including storage
change in the soil column to capture all terms of the water balance. All fluxes are scaled by the average annual rainfall.

The performance of all UHCC models is generally consistent, with only minor variations. Transpiration as a fraction
of the rainfall is highest for Tamale and lowest for Ukkel. The soil has no notable effect on transpiration, consistent with the
analysis above of the extreme values of Fig. 6. The scaled infiltration is nearly identical for Colombo and Tamale, and slightly
lower for Ukkel. By necessity, the scaled infiltration and overland flow add up to one because interception was not modelled.
The ratio of evaporation over transpiration is smallest for Colombo and largest for Ukkel, influenced by both the soil type and
the UHCC model. The semi-arid climate (Tamale) has the smallest fraction of the rainfall directed to the groundwater, the
monsoon climate (Colombo) the highest. The soil has a clear effect on this, but the effect of the UHCC is only noticeable for
UNSODA 2571. The scaled storage change is $-0.04$ to $-0.12$ for Pachepsky's soil, and within $\pm 0.02$ for the other soils.

### 3.5 Summary of the main findings

The JUV and KGV models are more robust than the ADV model, causing far fewer simulation failures. The effect of
the choice of UHCC model on numerically calculated fluxes is generally small. Even in a semi-arid climate, inclusion of the
film flow conductivity does not make a difference in the calculated groundwater recharge, or other fluxes. Differences in the
number of fitted parameters have only minimal impact on fluxes as a function of time, on various statistics of fluxes, as well
as on the soil the water balance. This offers opportunities to reduce the risk of overparameterization by choosing parsimonious
UHCC models and/or fixing some of the model parameters during parameter fitting.






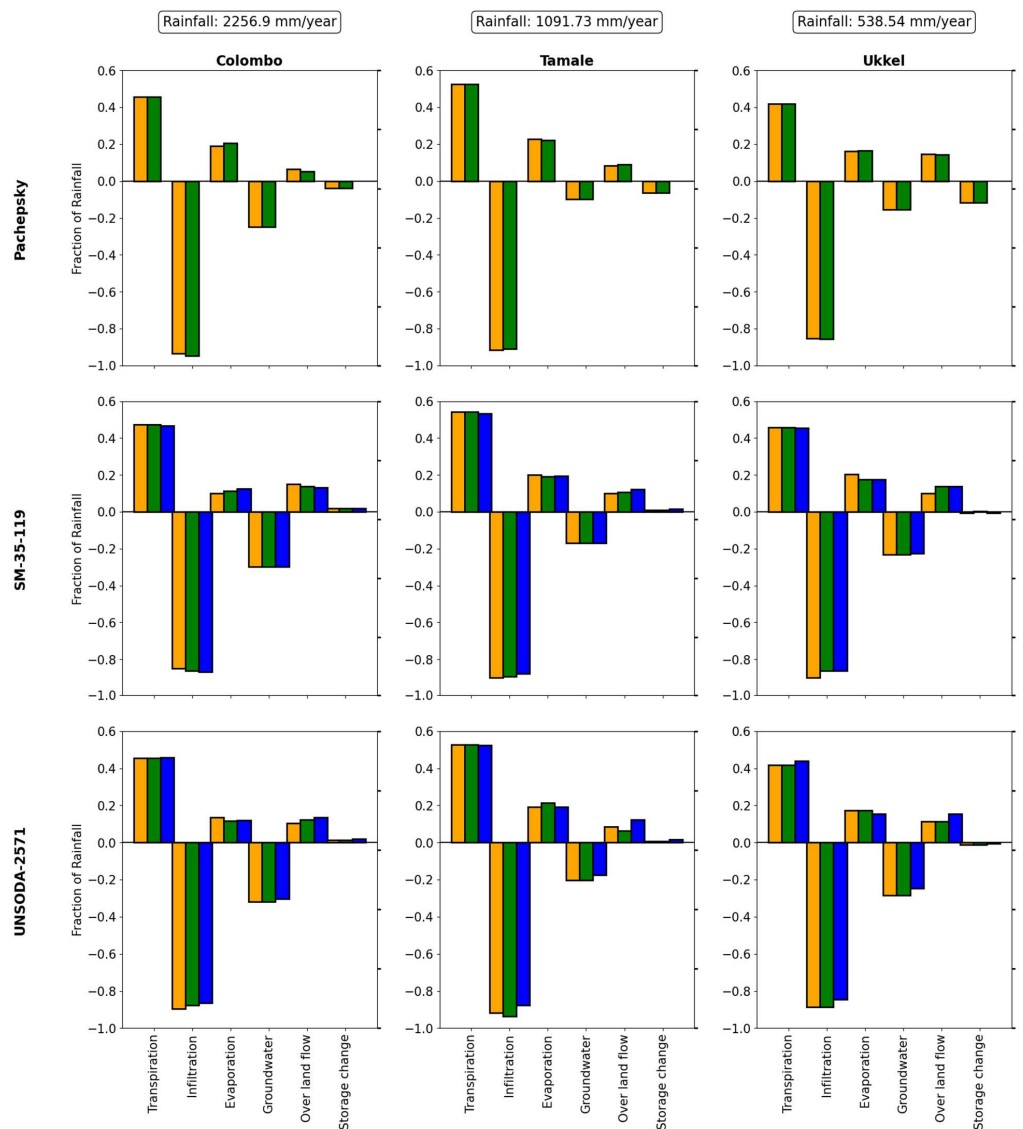

**Figure 9: Rainfall partitioning into infiltration, evaporation, transpiration, groundwater recharge, overland flow, and storage change across three climates and soils.**



**Appendix A: Graphs showing the fluxes through the vegetation, at the soil surface and at the bottom profile.**


The three UHCC models considered in this study were evaluated for three soils under three climates. De Rooij (2025) tested five different sets of fitting parameters for each UHCC model, but only used goodness-of-fit criteria to do so (the Root Mean Square Error – RMSE, and AICc). This appendix shows various simulated fluxes (with the noise filtered out as described in the main text) for all nine soil–climate combinations and all three UHCC models. For each UHCC model, Hydrus-1D was

run for all five sets of fitting parameters given in Section 2.1 of the main text. Below, 'Mualem' corresponds to fitting parameter set 4 in Section 2.1, 'Assouline' to set 3, 'Alpha only' to set 2, 'All fitted' to set 1, and 'SWRC fixed' to set 5.

Figures A1–A9 are for Pachepsky's soil (sandy loam), Figs. A10–A18 for SM–35–119 (silt), and Figs. A19–A27 for UNSODA 2571 (loamy sand). Within each set of nine figures per soil, nrs. 1–3 depict the actual transpiration, nrs. 4–6 the surface flux, i.e., infiltration (−) or actual evaporation (+), and nrs. 7–9 the flux across the bottom boundary. The latter is

always downward (−), consistent with the unit-gradient boundary condition there. Nrs. 1, 4, and 7 are for the monsoon climate (Colombo), nrs. 2, 5, and 8 for the semi-arid climate (Tamale), and nrs. 3, 6, and 9 for the temperate climate (Ukkel). The figure captions below simply list the soil-climate-flux configuration.

Occasionally, peaks appear in the time series, particularly at the start and end of the data. These peaks can sometimes reverse the sign of the flux to values that are not physically realistic. Such anomalies arise from the Savitzky-Golay filtering

algorithm, which smooths data by fitting a polynomial to a moving window of points. However, near the boundaries of the time series, the algorithm lacks sufficient data on one side of the window to perform a balanced fit, leading to edge effects. These edge effects manifest as artefacts or distortions that do not reflect the underlying physical processes. While these artefacts can be visually prominent, they are a known limitation of the filtering technique and can be safely disregarded in the analysis.

The performance of Hydrus in terms of number of successful runs is treated in the main text. In this appendix, we focus on the effect of the choice of the UHCC model and the fitting parameter set on the simulation results. We contrast these with the effects of the weather and the soil type.

All model runs capture inter-annual variations in the weather conditions well (see the rainfall records in Fig. 5 in the main text). The actual transpiration in the monsoon climate has strong day-to-day variations (Figs. A1, 10, 19). For the semi-

arid climate, such variations are absent (Figs. A2, 11, 20), and for the temperate climate (Figs. A3, 12, 21), they are limited to the season without precipitation deficit. This probably reflects the damping effect of soil-limited transpiration. If the water-availability is not limiting, the actual transpiration will be close to its potential values, and daily fluctuations are not damped. The transpiration is affected by the UHCC models and the fitting parameter set for Pachepsky's soil only (Fig. A3).

The surface flux (infiltration or actual evaporation) is hardly affected by soil type, UHCC model, or fitting parameter

set (Figs. A4, 13, 22 for the monsoon climate, Figs. A5, 14, 23 for the semi-arid climate, and Figs. A6, 15, 24 for the temperate climate). Apparently, the surface flux is almost exclusively determined by the weather.



The bottom boundary flux is expected to be most strongly affected by the soil hydraulic properties, because the infiltrated water has to travel through the entire profile before it can exit at the bottom. For the monsoon climate, all soils have alternating periods with zero and high bottom flux, with rapid transitions (Figs. A7, 16, 25). The effect of the choice of UHCC model or fitting parameter set is negligible, but there is a distinct, although not very large, influence of the soil type.

For the semi-arid climate, SM–35–119 (silt) and UNSODA 2571 (loamy sand) have a non-zero bottom flux in nine out of ten years (Figs. A17, 26). The peak discharge rate varies by an order of magnitude between these years. Especially in dry years, the loamy sand generates more bottom flux than the silt. The effect of the UHCC model and the fitting parameter set is clearly distinguishable, but not very large. Pachepsky's sandy loam behaves much differently (Fig. A8). Years 3 through 5 have nearly no bottom flux for KGV and JUV (Hydrus-1D crashed for ADV). But if fitting parameter sets 1 (All fitted) or 3 (Assouline) are chosen for JUV, nine out of ten years have a bottom flux, and the peaks of all but that of the wettest year are twice as high as those for KGV. Still, these fluxes are smaller than those of the other two soils.

The bottom flux for ADV in Pachepsky's sandy loam in the temperate climate has a much stronger seasonality than for KGV and JUV for all fitting parameter sets for which ADV converged, with one exception for JUV. For the silt (SM–35–119), there is a clear but not very large effect of the fitting parameter set on the bottom flux, but not of the UHCC model. For the loamy sand (UNSODA 2571), the UHCC and the fitting parameter set both affect the bottom flux to a limited degree.



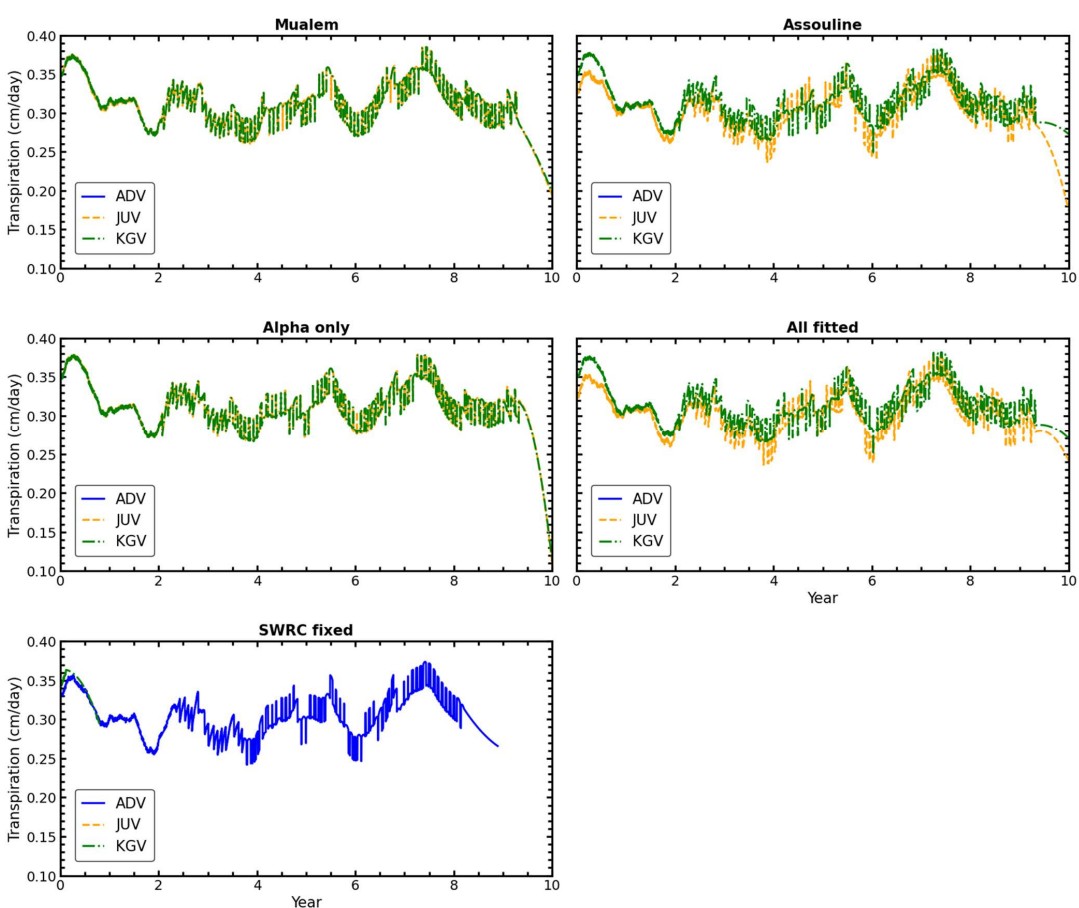

**Figure A1: Soil: Pachepsky (sandy loam). Climate: monsoon. Flux: Actual transpiration.**



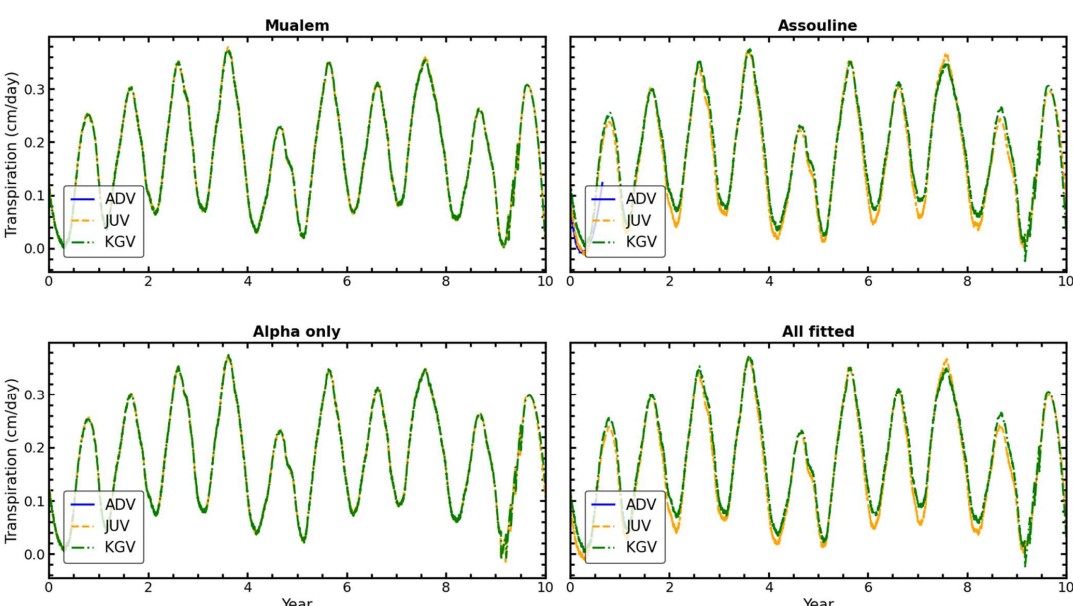


**Figure A2: Soil: Pachepsky (sandy loam). Climate: semi-arid. Flux: Actual transpiration.**



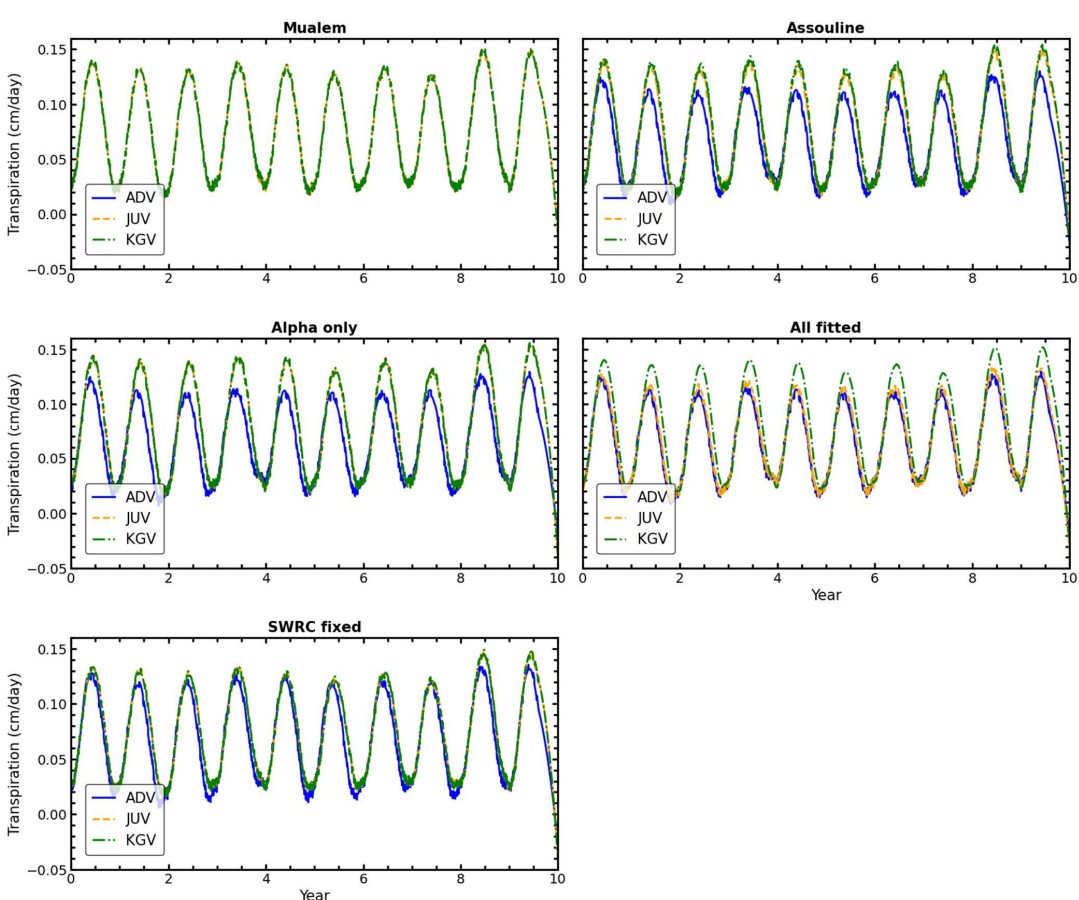

**Figure A3: Soil: Pachepsky (sandy loam). Climate: temperate. Flux: Actual transpiration.**




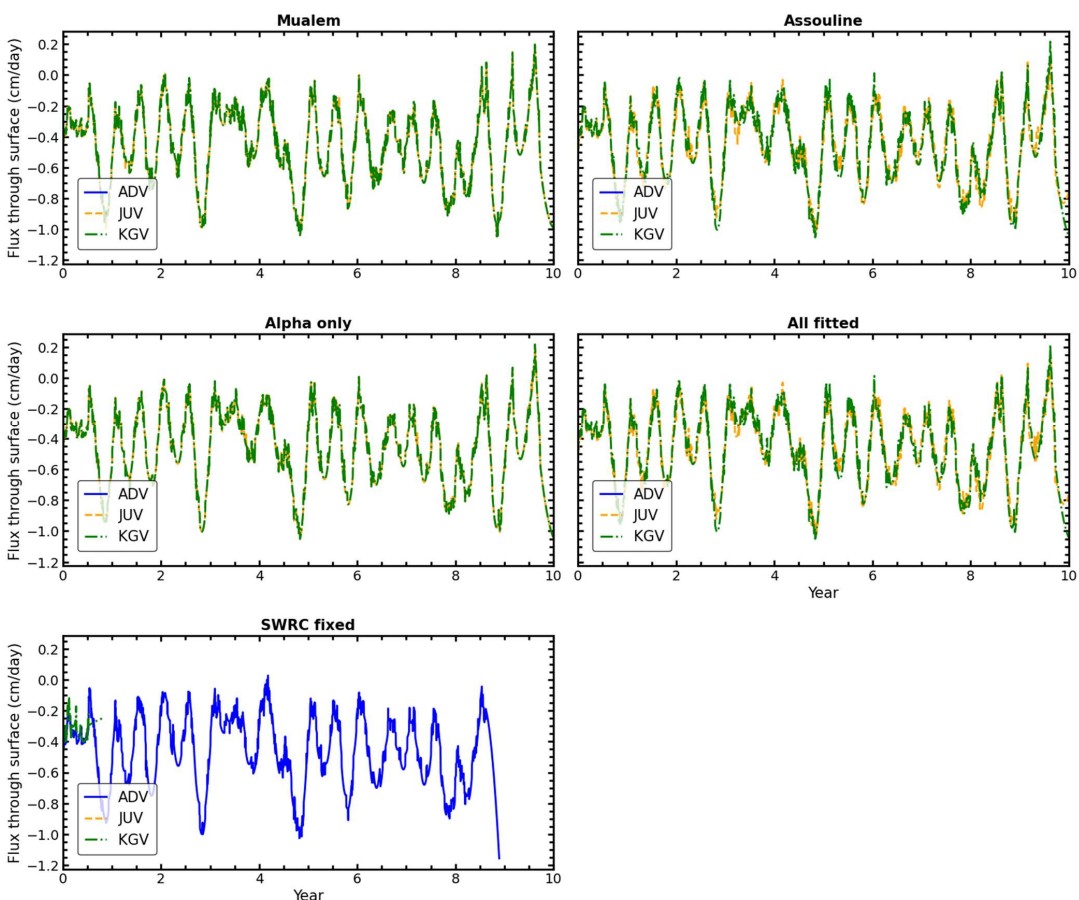

**Figure A4: Soil: Pachepsky (sandy loam). Climate: monsoon. Flux: Surface flux.**





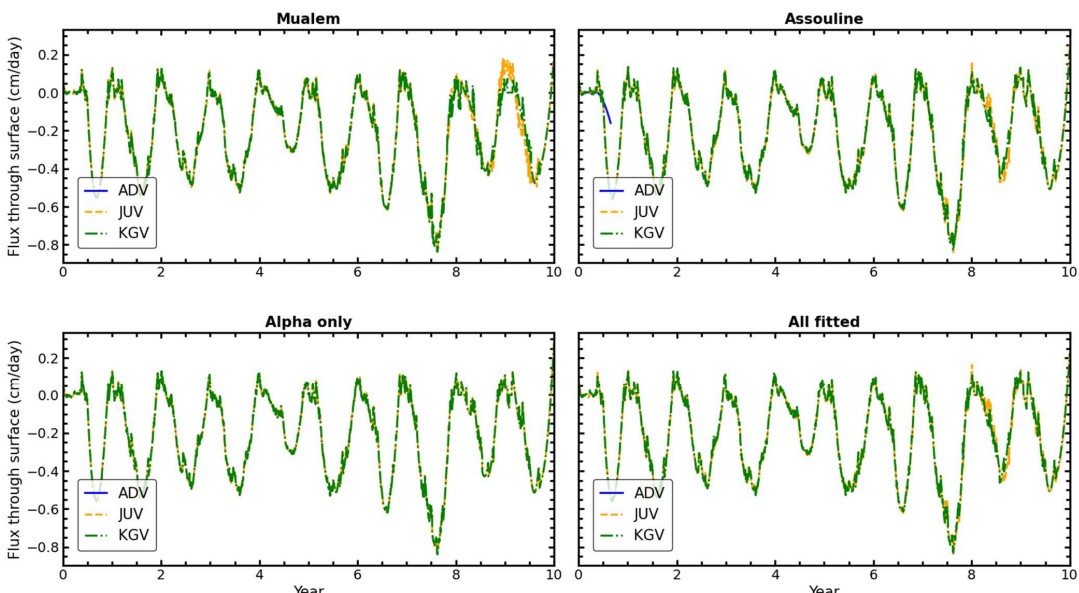

**Figure A5: Soil: Pachepsky (sandy loam). Climate: semi-arid. Flux: Surface flux**




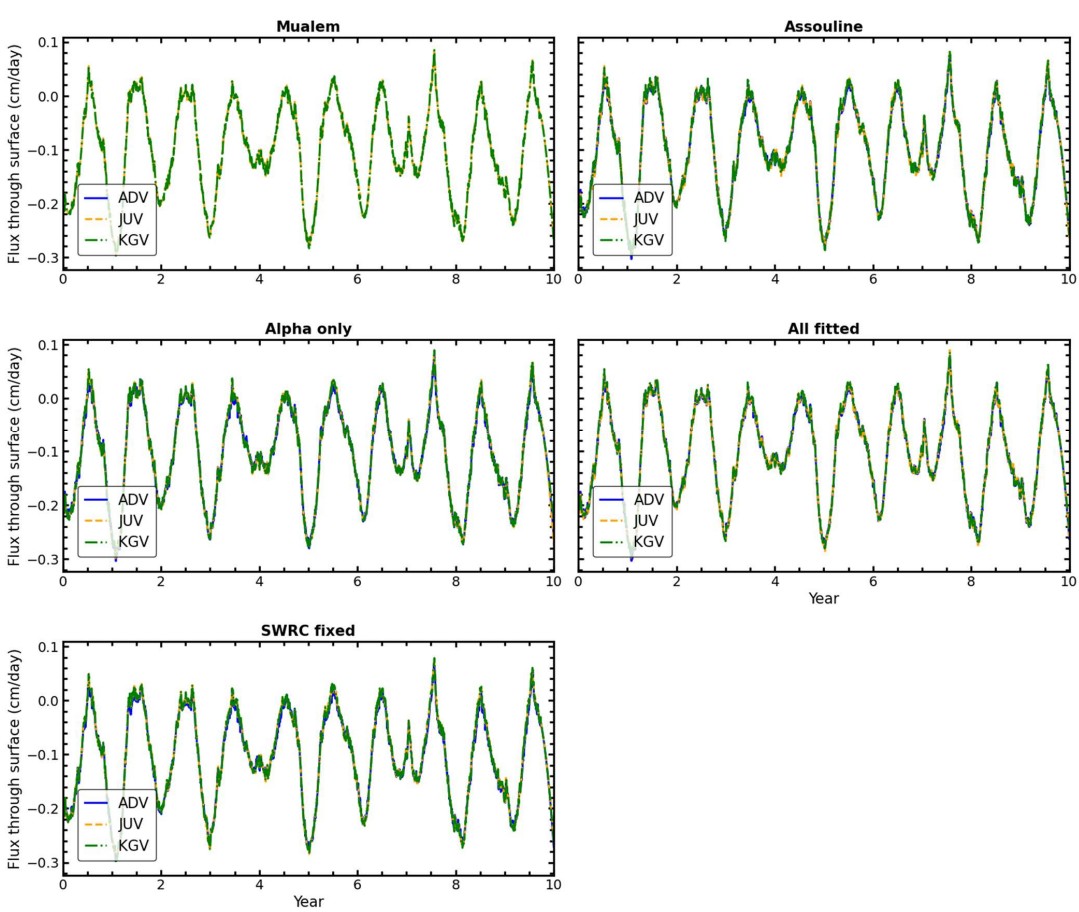

Figure A6: Soil: Pachepsky (sandy loam). Climate: temperate. Flux: Surface flux.



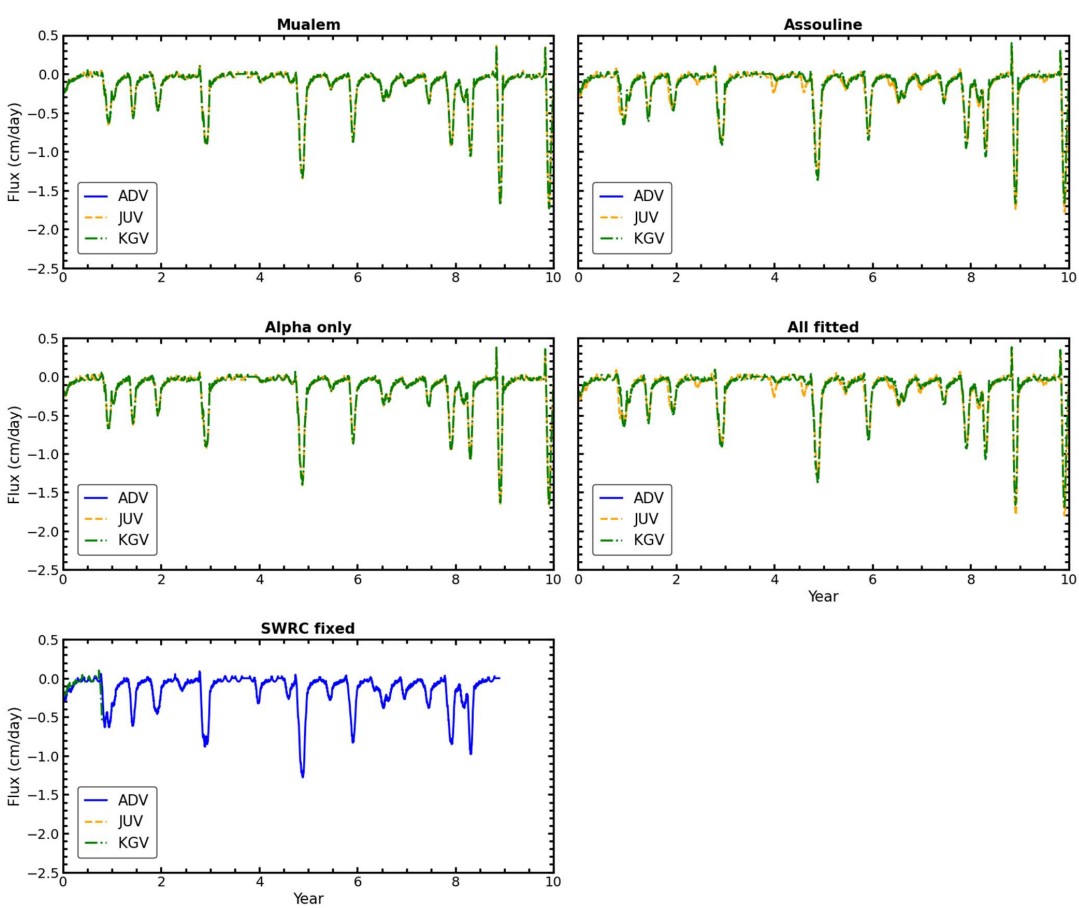

**Figure A7: Soil: Pachepsky (sandy loam). Climate: monsoon. Flux: Bottom boundary flux.**





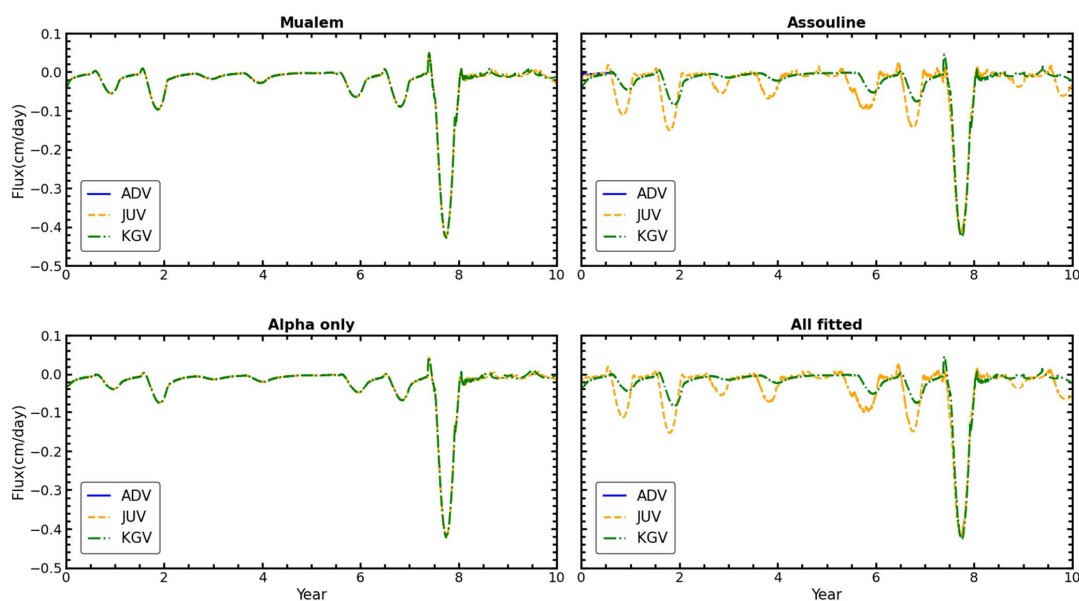


**Figure A8: Soil: Pachepsky (sandy loam). Climate: semi-arid. Flux: Bottom boundary flux.**




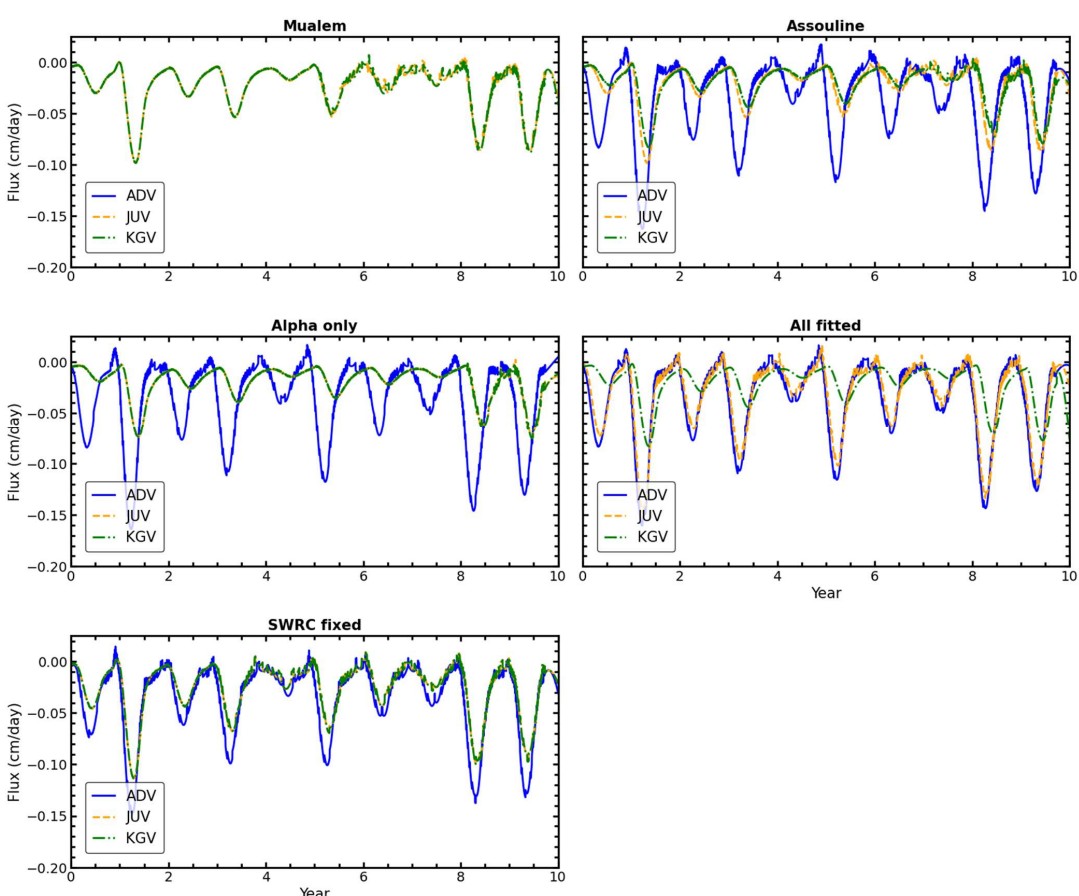

**Figure A9: Soil: Pachepsky (sandy loam). Climate: temperate. Flux: Bottom boundary flux.**





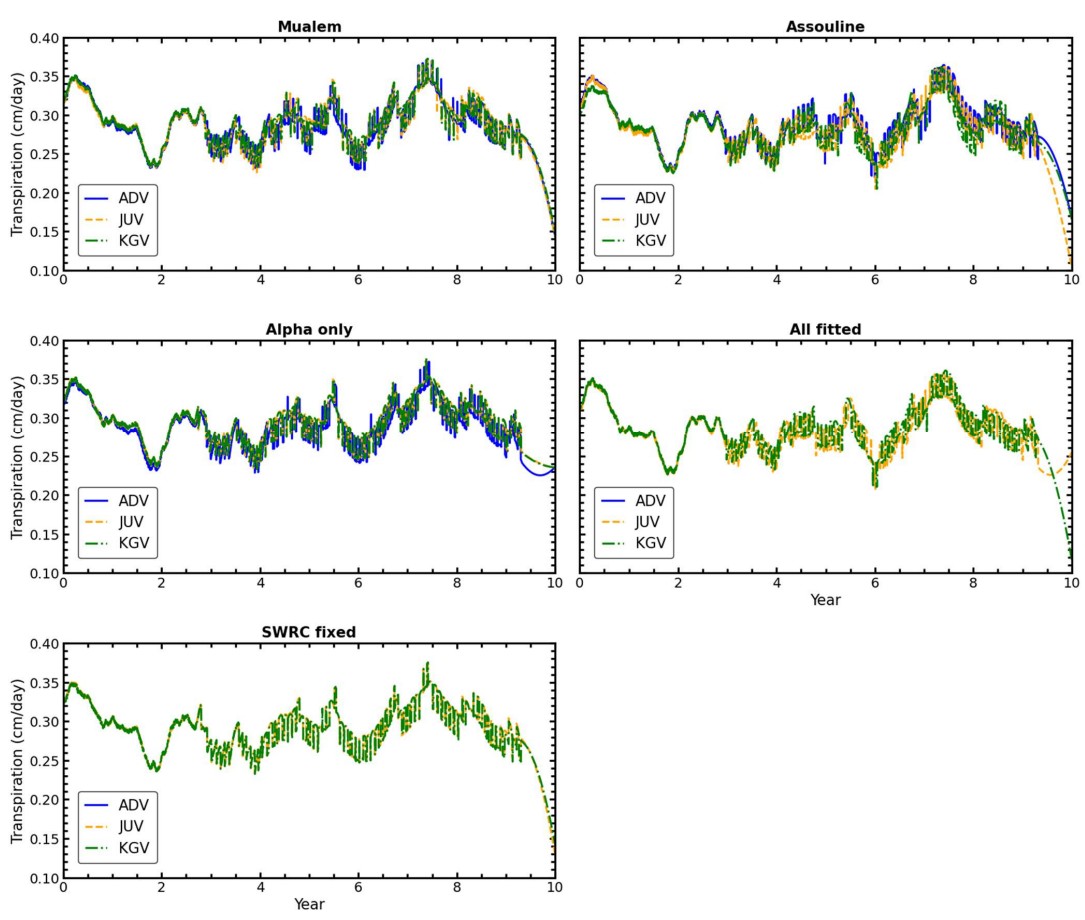

**Figure A10: Soil: SM–35–119 (silt). Climate: monsoon. Flux: Actual transpiration.**






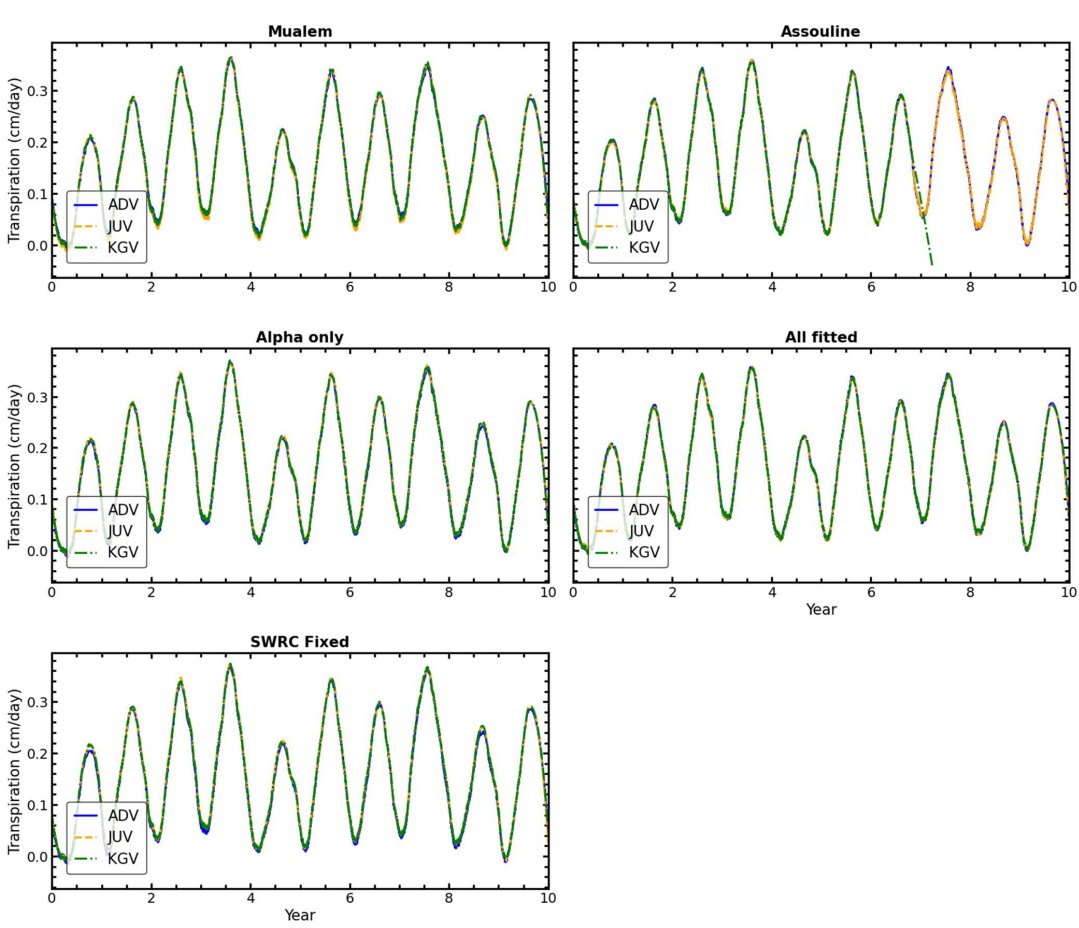

**Figure A11: Soil: SM–35–119 (silt). Climate: semi-arid. Flux: Actual transpiration.**





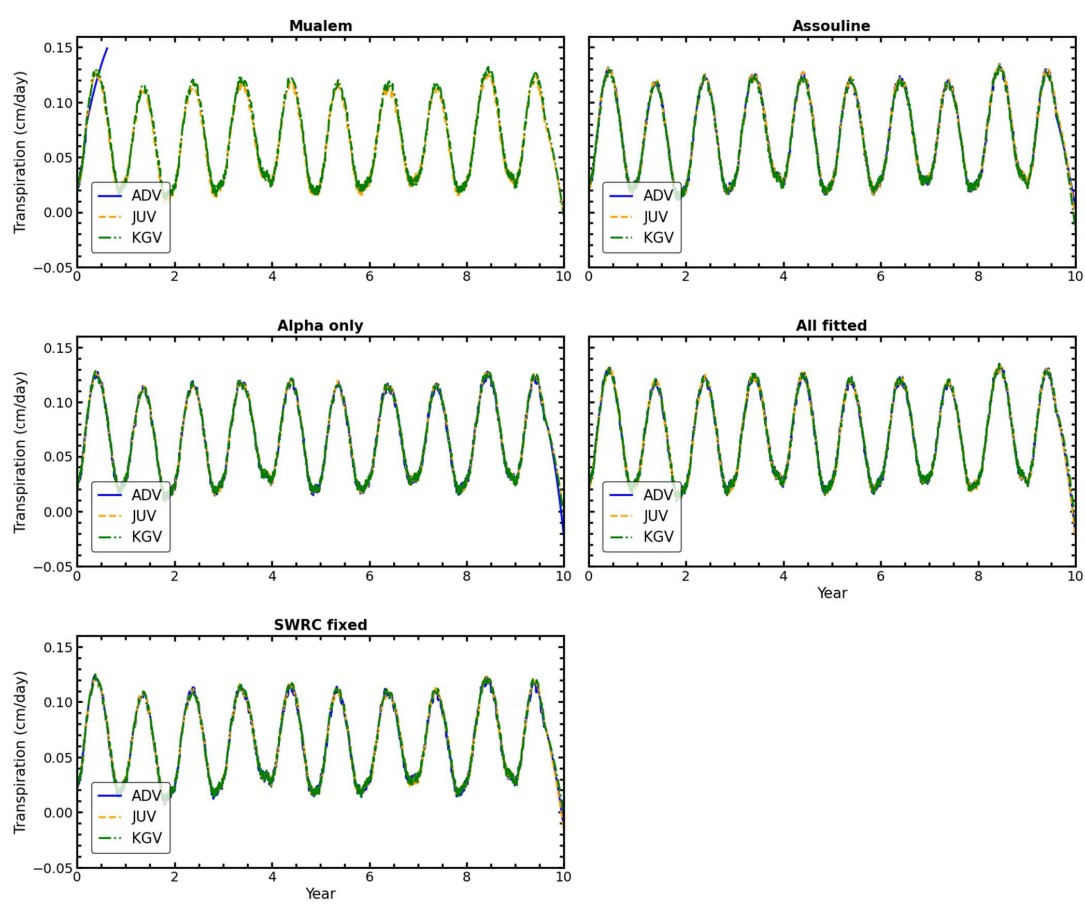


**Figure A12: Soil: SM–35–119 (silt). Climate: temperate. Flux: Actual transpiration.**



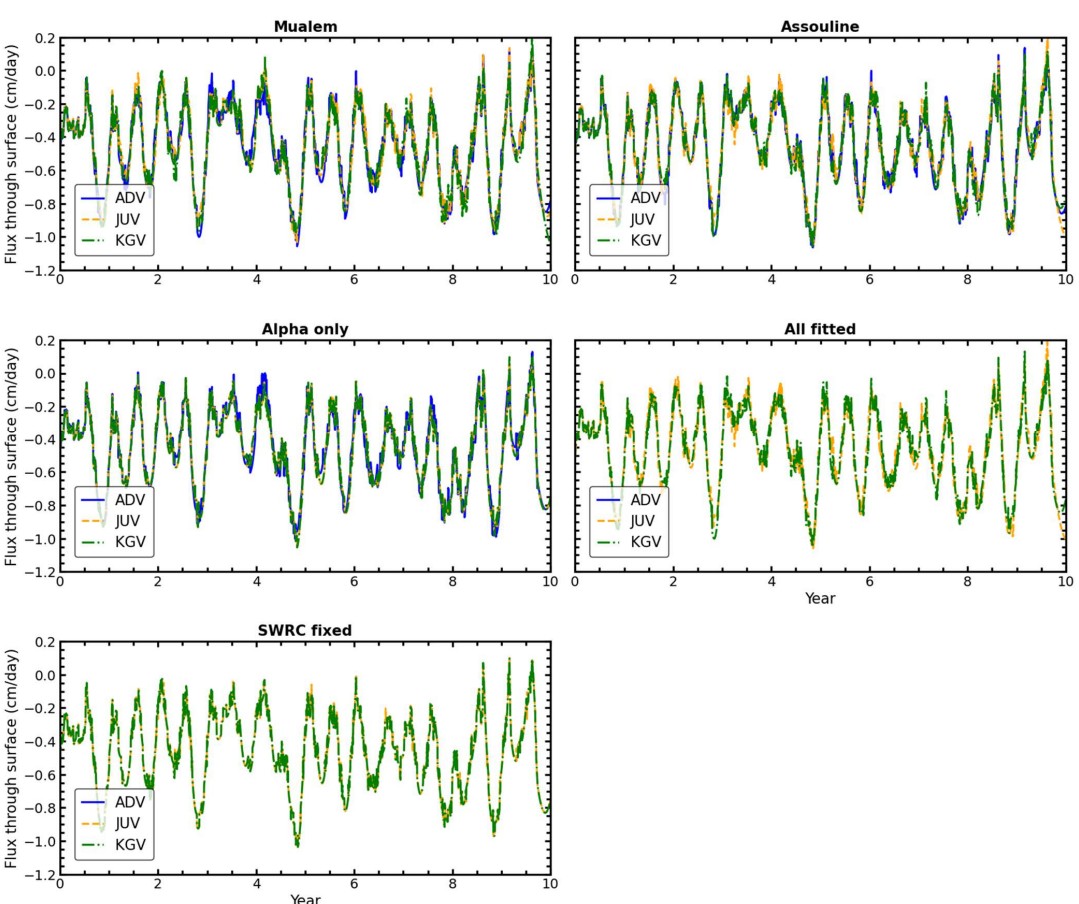

**Figure A13: Soil: SM–35–119 (silt). Climate: monsoon. Flux: Surface flux.**




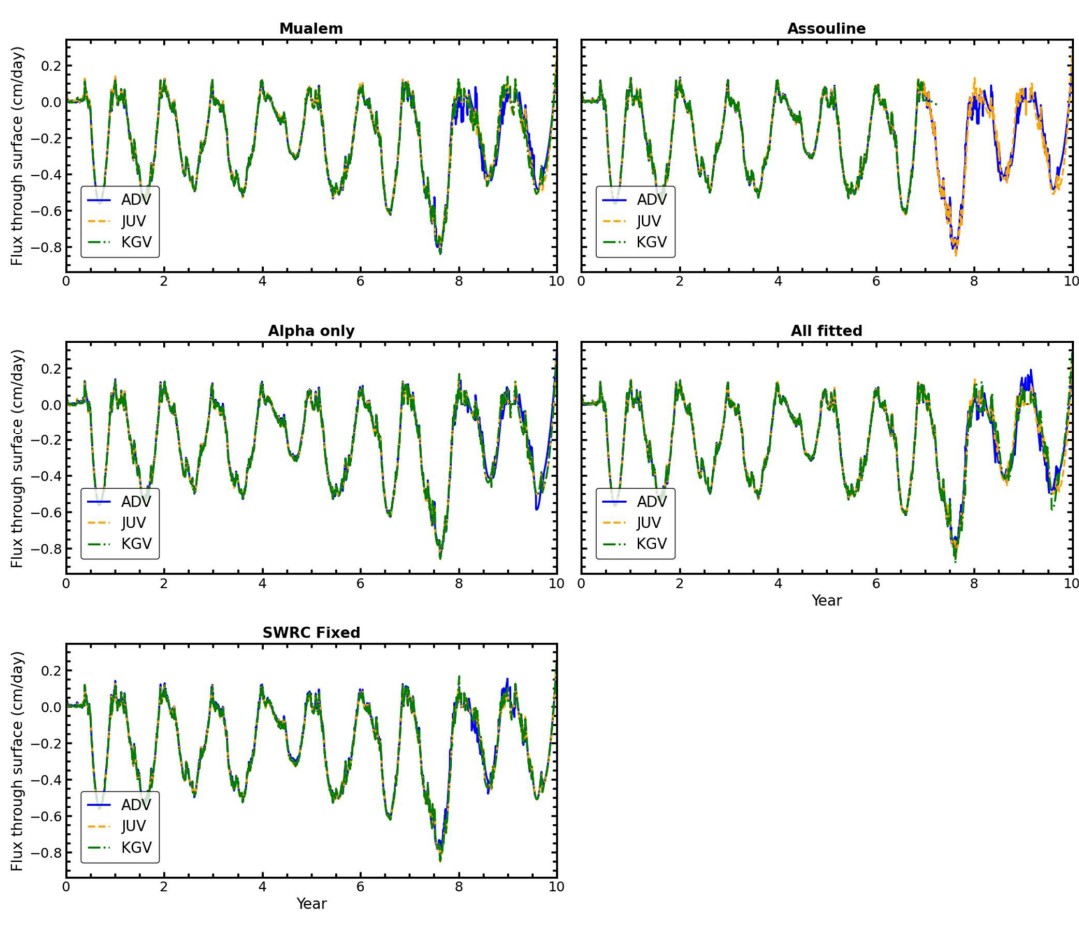


Figure A14: Soil: SM–35–119 (silt). Climate: semi-arid. Flux: Surface flux.





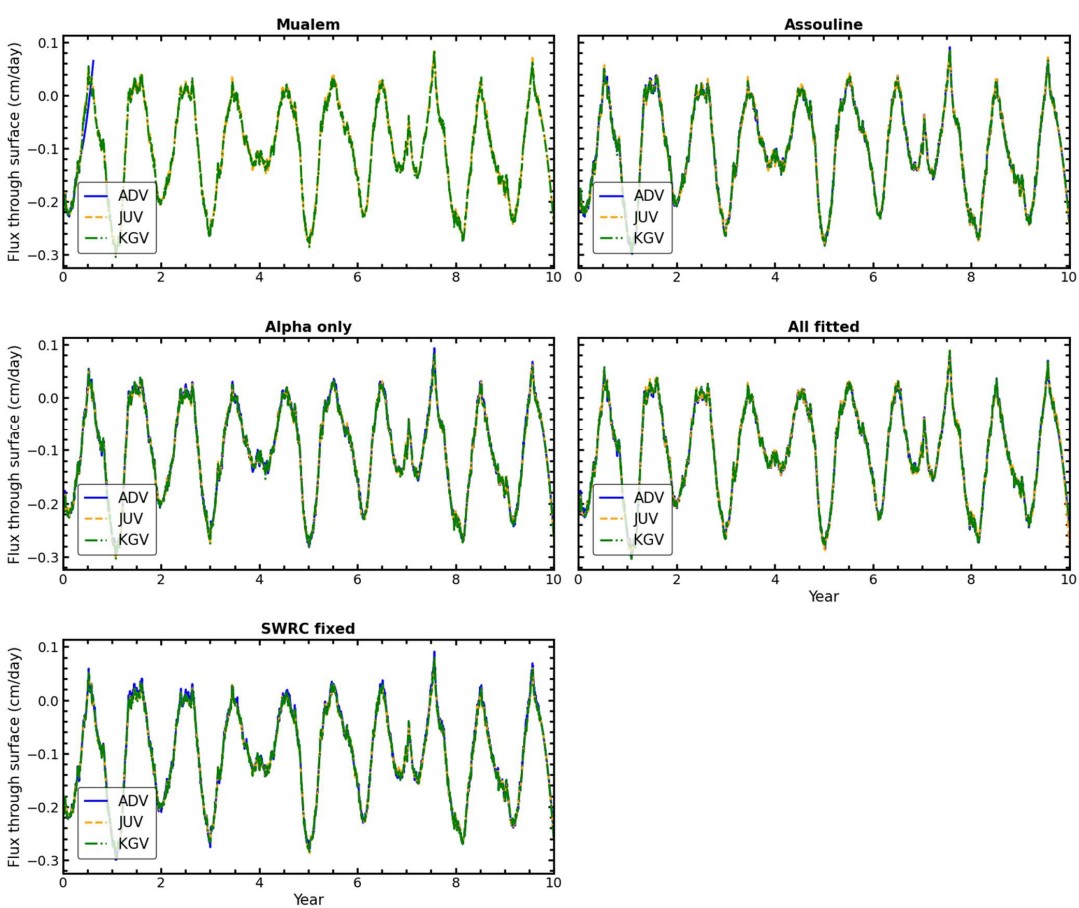

**Figure A15: Soil: SM–35–119 (silt). Climate: temperate. Flux: Surface flux.**






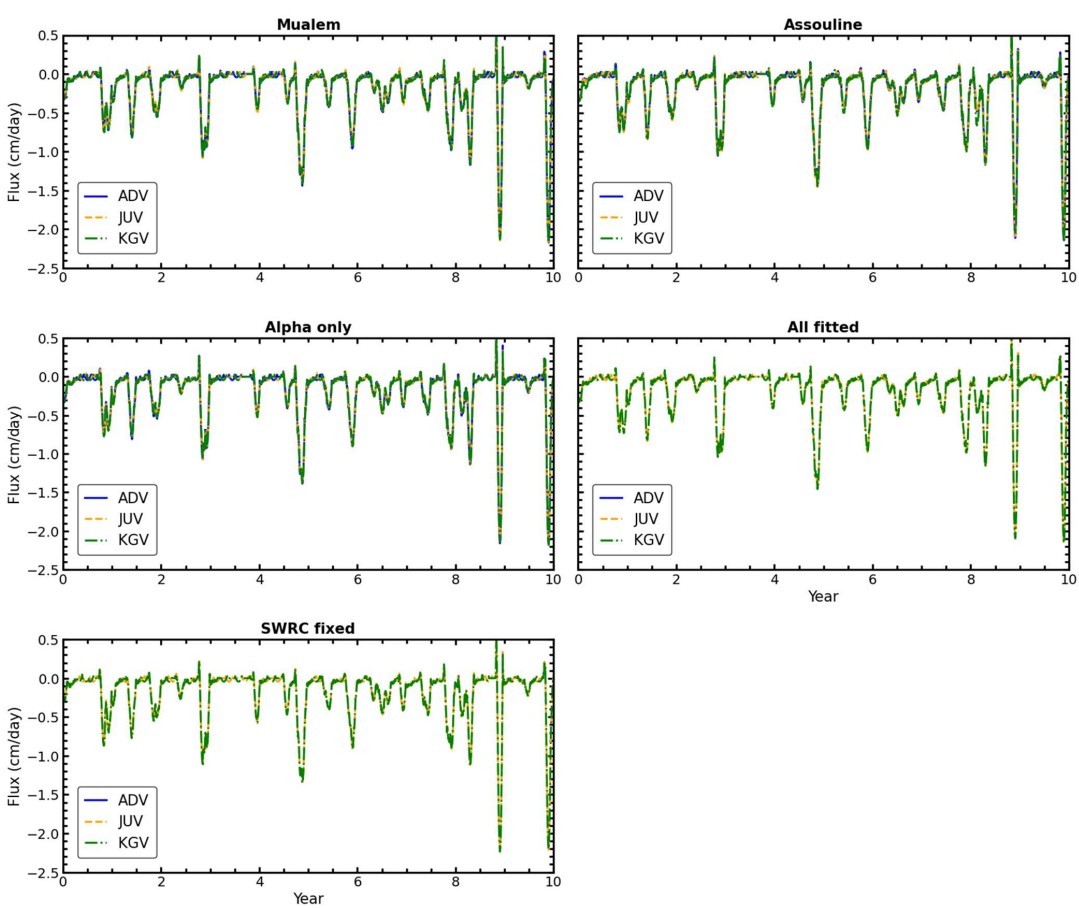

**Figure A16: Soil: SM–35–119 (silt). Climate: monsoon. Flux: Bottom boundary flux.**



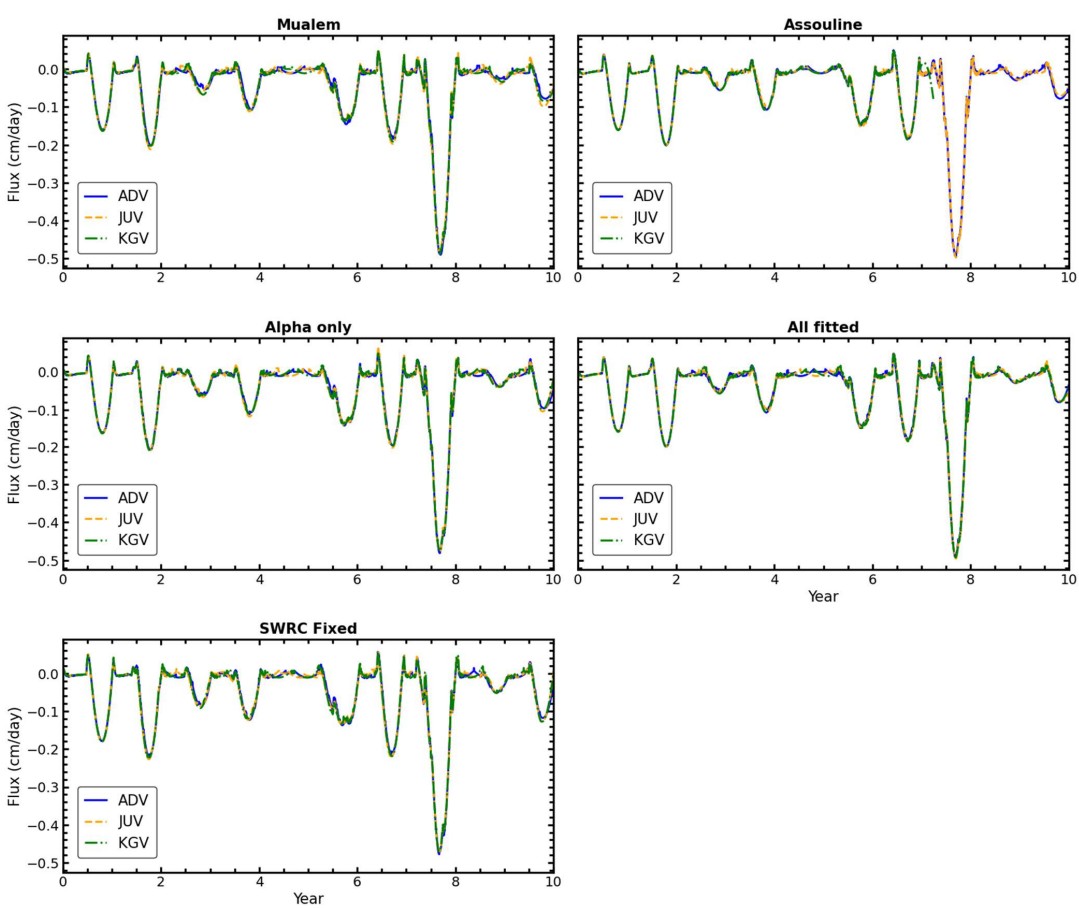

**Figure A17: Soil: SM–35–119 (silt). Climate: semi-arid. Flux: Bottom boundary flux.**





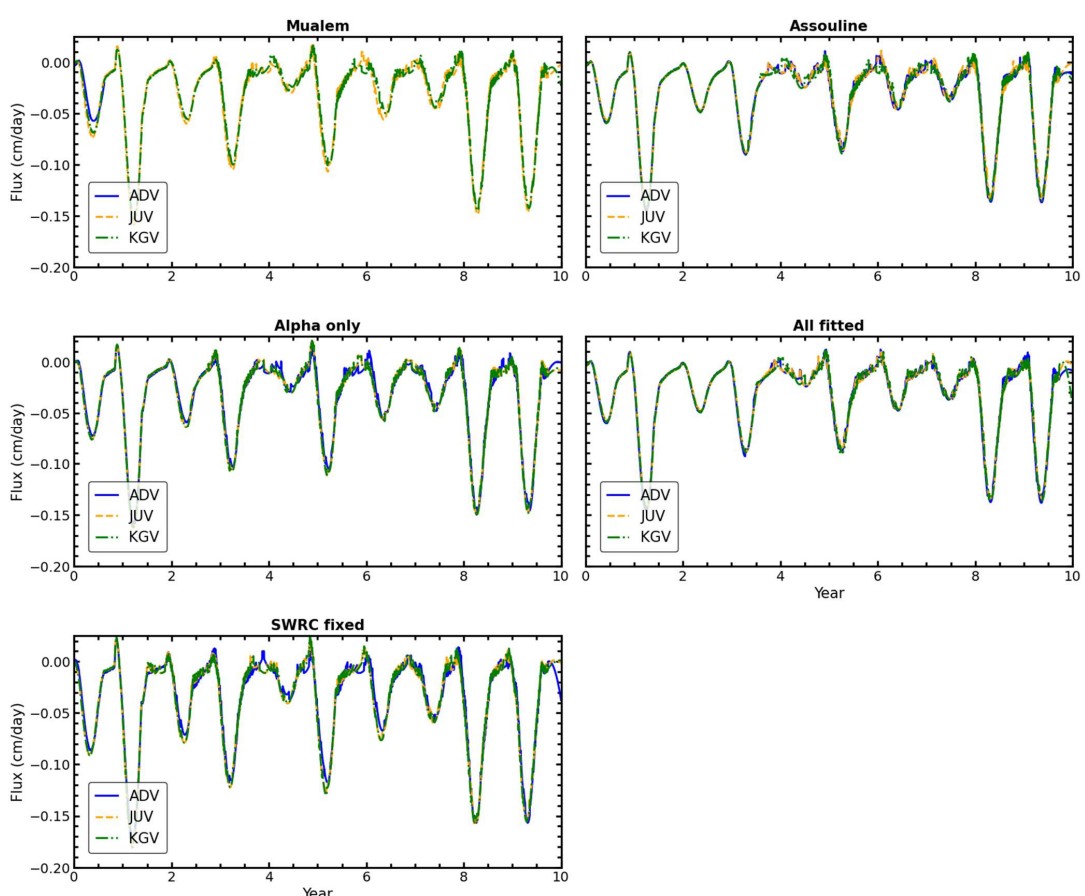

**Figure A18:** Soil: SM–35–119 (silt). Climate: temperate. Flux: Bottom boundary flux.



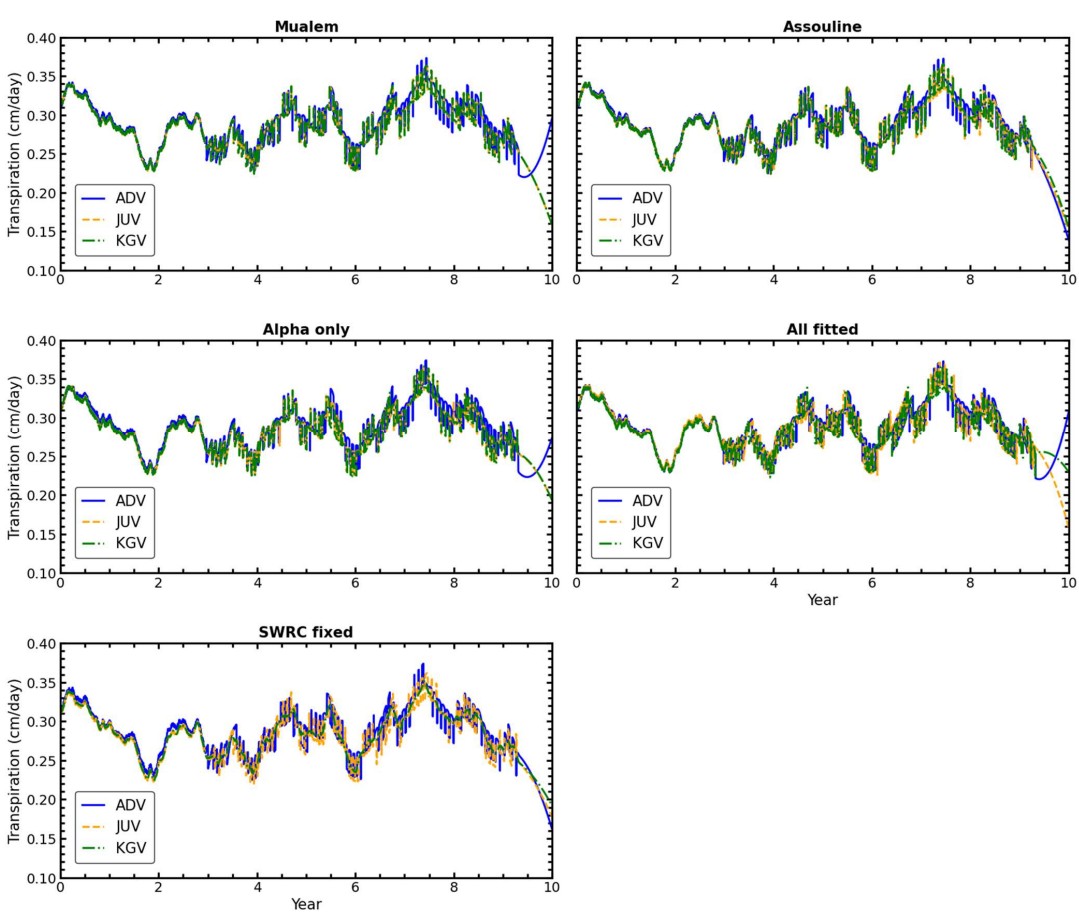

**Figure A19: Soil: UNSODA 2571 (loamy sand). Climate: monsoon. Flux: Actual transpiration.**



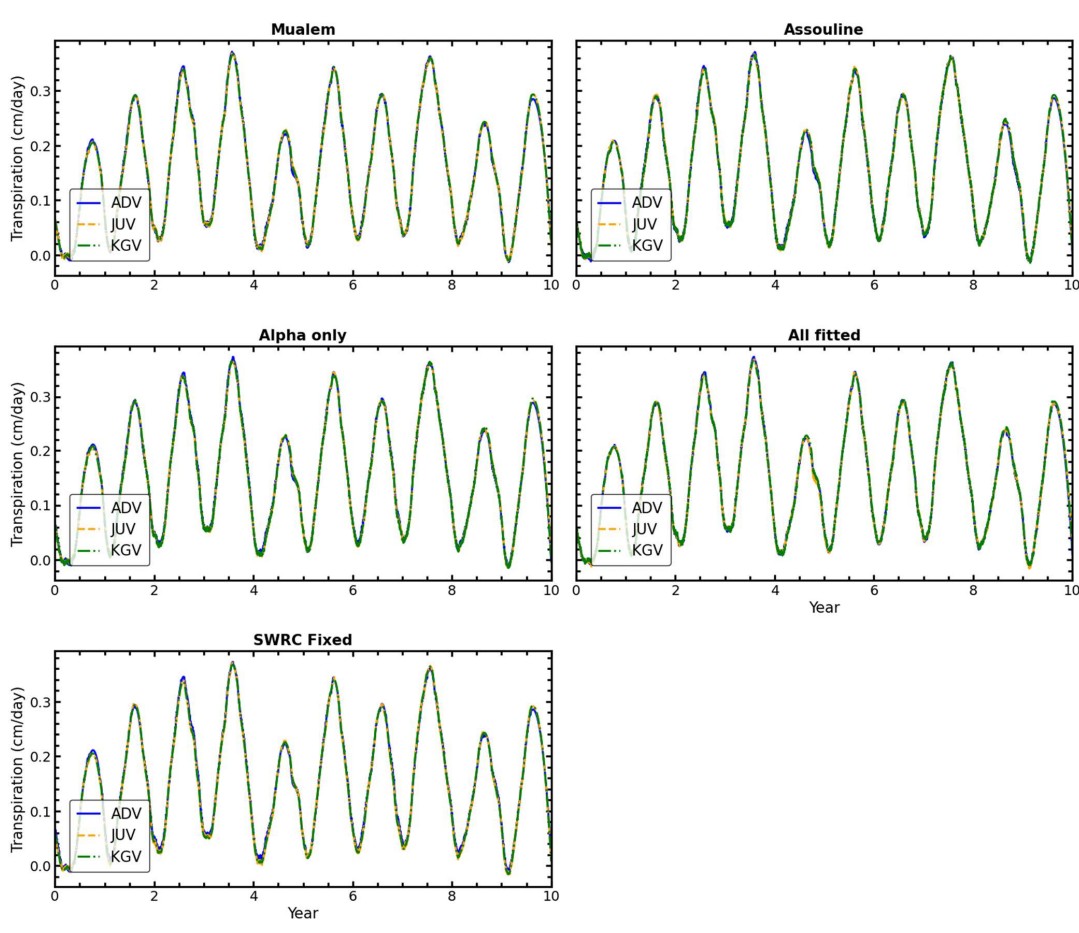


**Figure A20: Soil: UNSODA 2571 (loamy sand). Climate: semi-arid. Flux: Actual transpiration.**



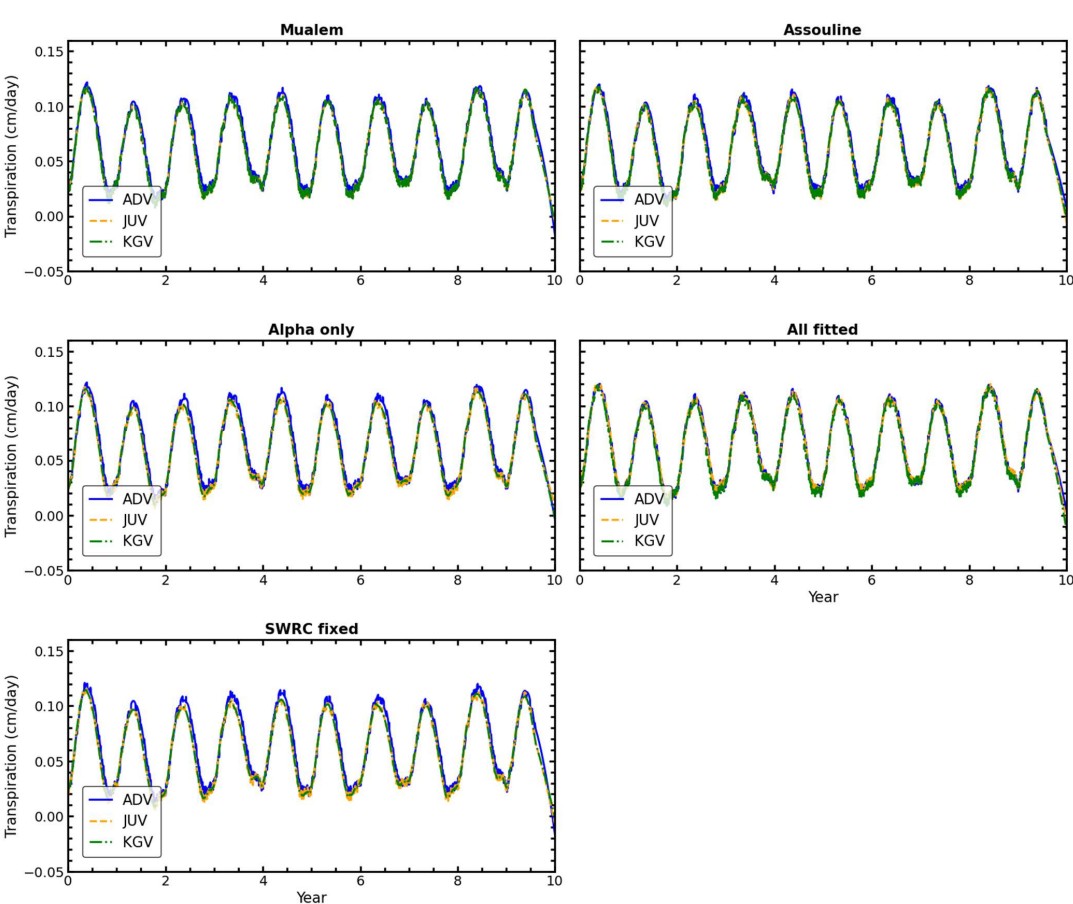

**Figure A21: Soil: UNSODA 2571 (loamy sand). Climate: temperate. Flux: Actual transpiration.**



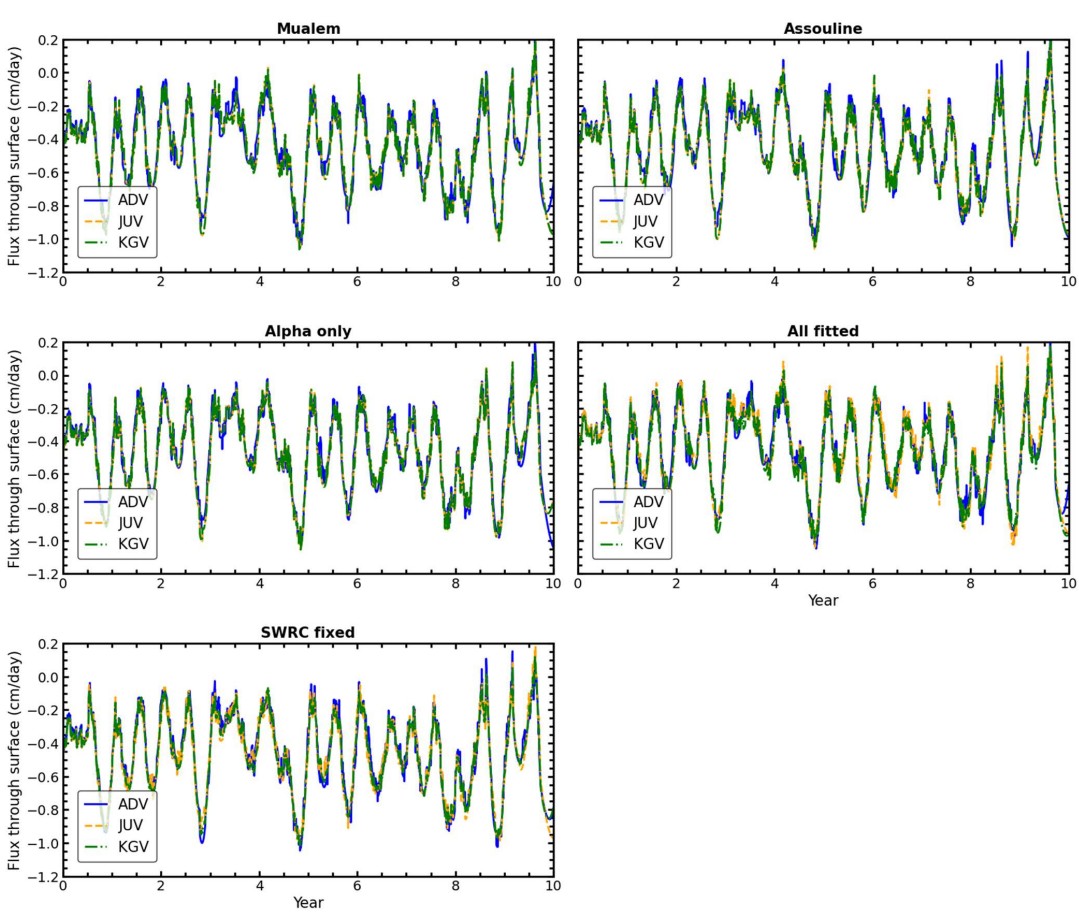


**Figure A22: Soil: UNSODA 2571 (loamy sand). Climate: monsoon. Flux: Surface flux.**



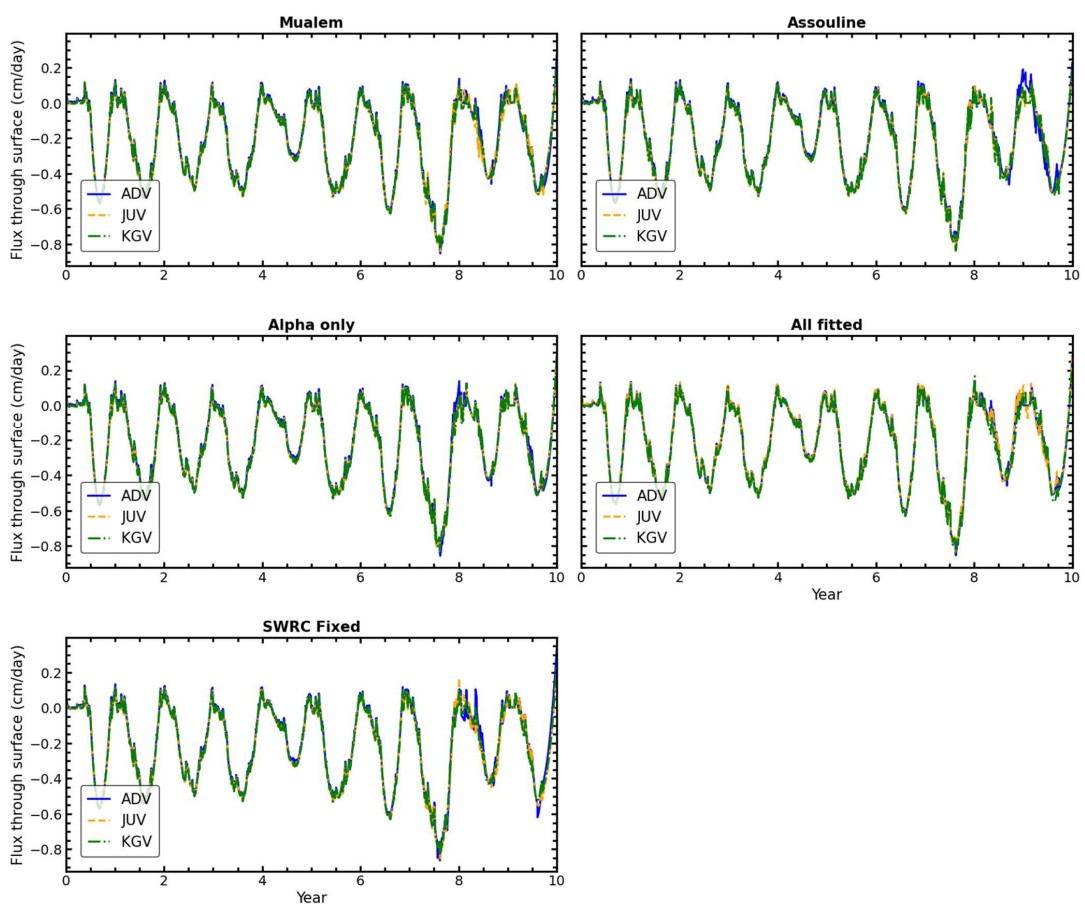

**Figure A23: Soil: UNSODA 2571 (loamy sand). Climate: semi-arid. Flux: Surface flux.**




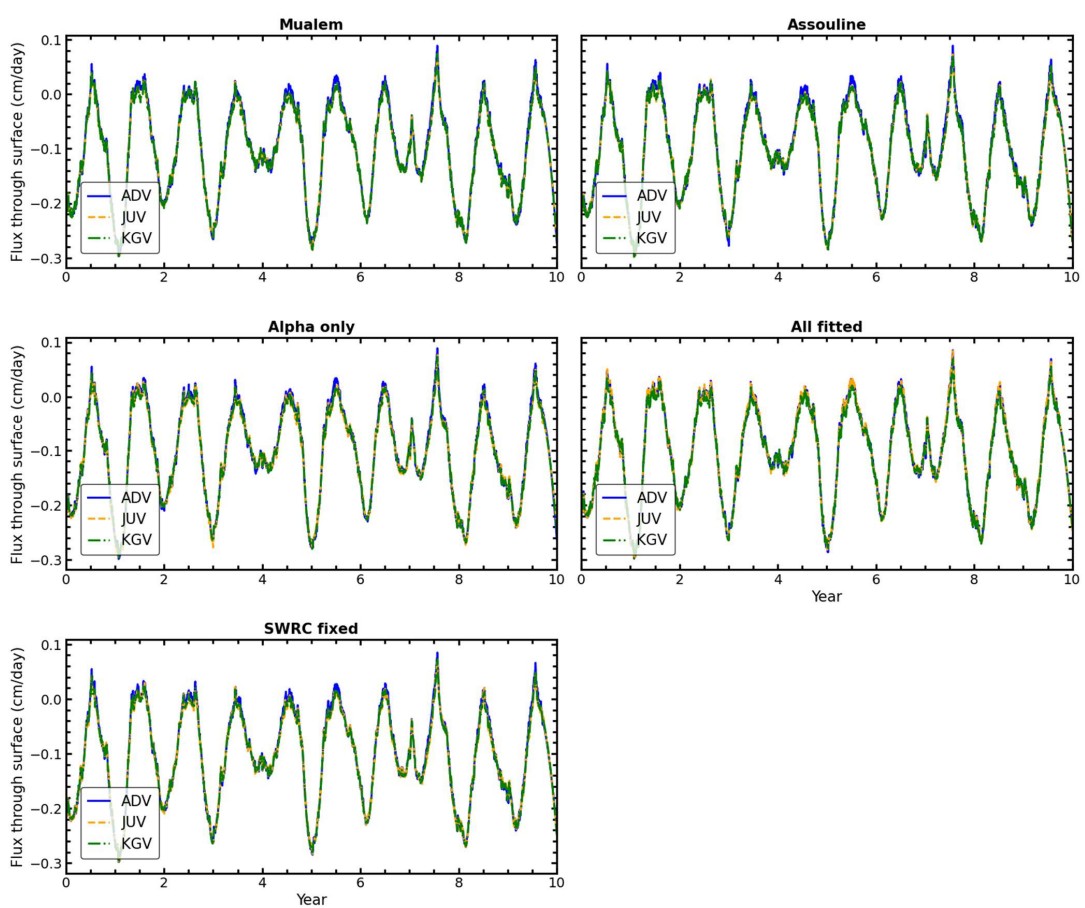

**Figure A24: Soil: UNSODA 2571 (loamy sand). Climate: temperate. Flux: Surface flux.**



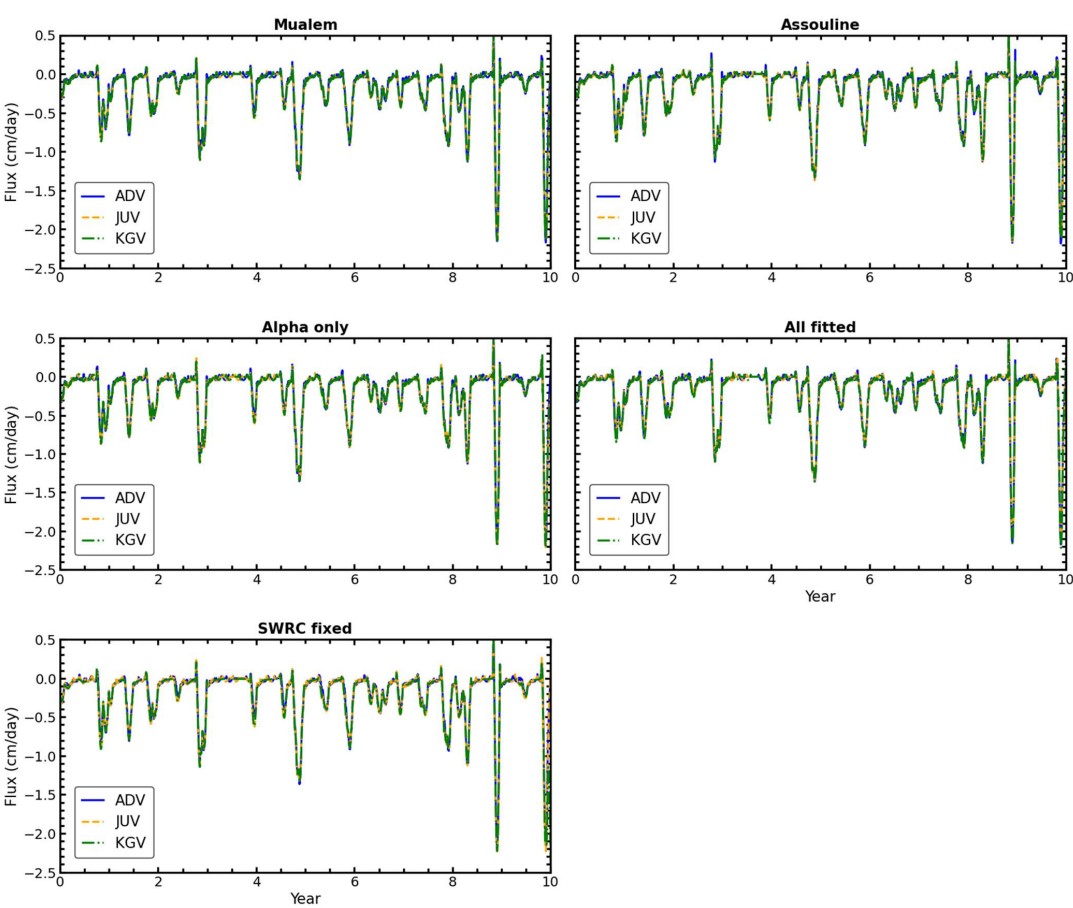

**Figure A25: Soil: UNSODA 2571 (loamy sand). Climate: monsoon. Flux: Bottom boundary flux.**





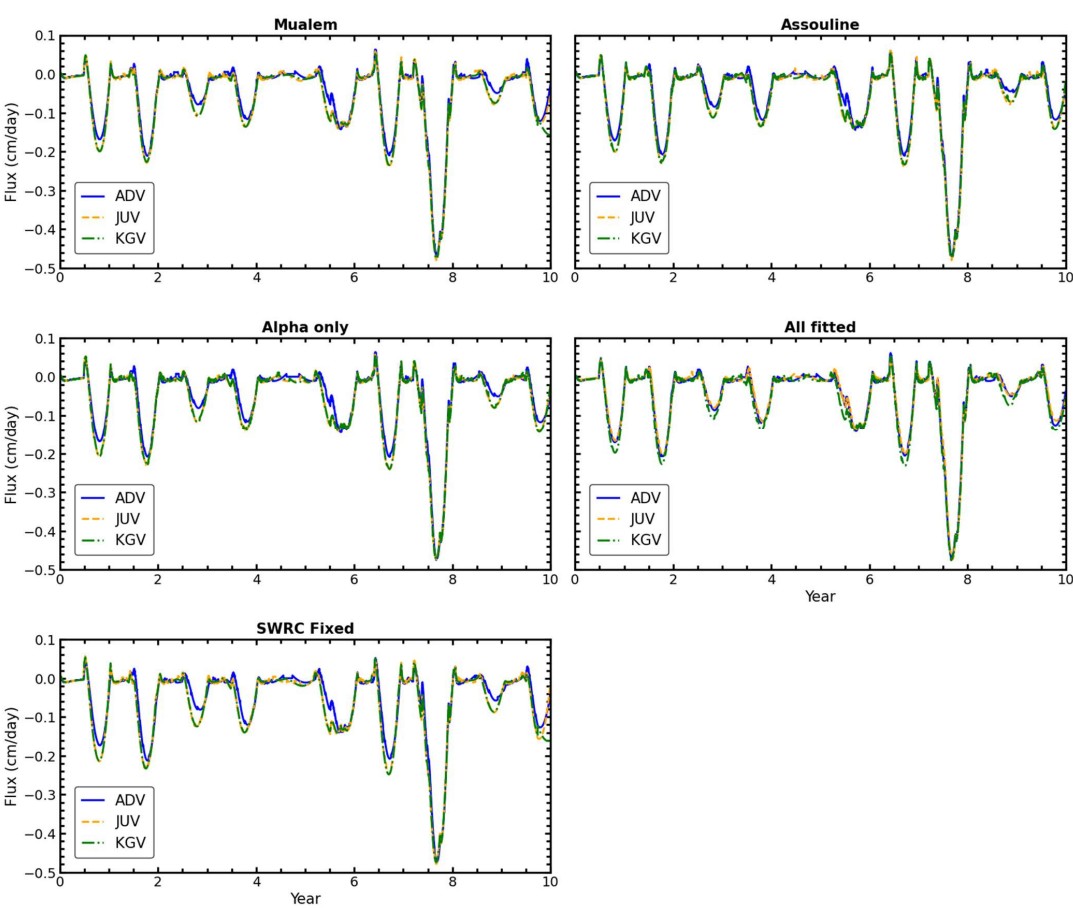

**Figure A26: Soil: UNSODA 2571 (loamy sand). Climate: semi-arid. Flux: Bottom boundary flux.**





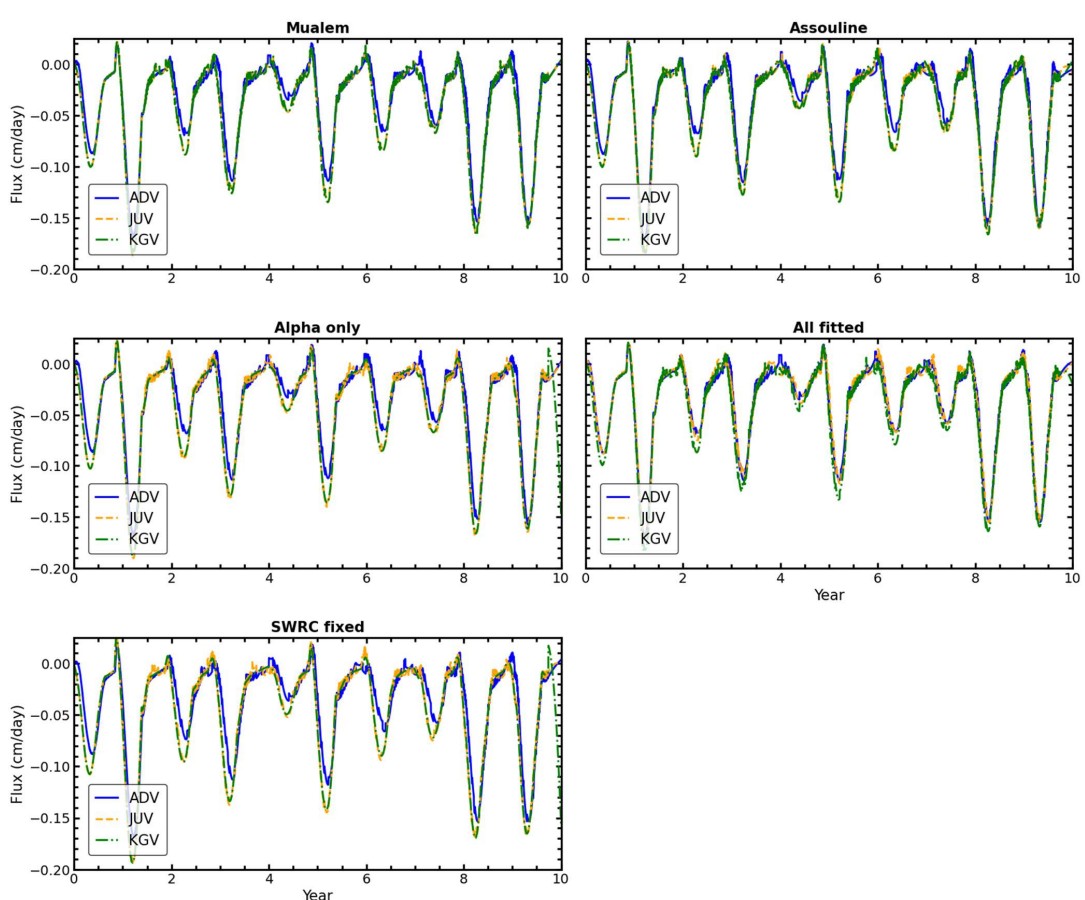

**Figure A27: Soil: UNSODA 2571 (loamy sand). Climate: temperate. Flux: Bottom boundary flux.**


## Appendix B: List of abbreviations

| | |
|---|---|
| ADV | Additive model of the UHCC |
| AICc | Akaike's Information Criterion for small sample sizes |
| JUV | Junction model of the UHCC |



| 420 | KGV | Kosugi's model of the UHCC |
| | pF | Log$(-h)$, with $h$ in cm H$_2$O |
| | RMSE | Root Mean Square Error |
| | SD | Standard deviation |
| | SWRC | Soil water retention curve |
| 425 | UHCC | Unsaturated soil hydraulic conductivity curve |
| | UNSODA | Unsaturated Soil Hydraulic Database |

**Data availability**

The processed output from Hydrus-1D and some input can be downloaded at https://zenodo.org/records/14753321 (doi: https://doi.org/10.5281/zenodo.14753321)

**Author contribution**

GdR conceptualized the study, selected the test soils and provided the fitted parameter values. GdR and AN designed the set of simulations to generate simulated data. AN prepared all input, ran the simulations, processed the output, and created the output visualizations. AN and GdR analyzed the output. GdR and AN wrote the draft version of the paper and reviewed and edited the final version.

**Competing interests**

GdR is a member of the editorial board of HESS.

**Financial support**

This research was partially funded by the Forest Climate Fund (Waldklimafonds - WKF) grant no. 2218W57B4.

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
