# Peer review of "Evaluating Unsaturated Hydraulic Conductivity Models for Diverse Soils and Climates: A Functional Comparison of Additive, Junction, and Kosugi Parameterizations"

_EGUsphere, 2025_

## Author Comment (AC1)

Comments of reviewer 1

Comments in *italics*, reply in regular font.

*The proposed study compares three different models for the unsaturated hydraulic conductivity curve (UHCC), namely the Kosogi model, the additive and junction model within a numerical exercise. Each of these models is fitted to data obtained for three different soils, using 5 different variations of free fitting parameters, and these are used to simulated the water balance of a 2 m homogeneous soil column for a temperate, Monsoon and semi-arid climate using weather data from sites in Ukkel in Belgium, Tamale in Ghana and Colombo in Sri Lanka. The comparison reveals that most simulated fluxes show a stronger sensitivity to climate, while the drainage flux in 2m shows the strongest sensitivity to changes in soil hydraulic properties.*

Reply:
The summary is essentially correct.

*I fully agree with the authors, that the ultimate purpose of UHCC is to be used in simulations, and I would highly value a comparison of those different models against independent data obtained in different real-world settings using a model of appropriate complexity.*

Reply:
  Independent, real-world data of soil water fluxes across the soil surface and below a certain depth, in combination with storage changes, can only be measured by suitably instrumented weighing lysimeters, the construction of which costs in the order of $10^5$ euros/dollars. The subsequent operation, ideally for multiple years, requires a staff of several full-time technicians to operate. Both reviewers appear to require from us to devote a substantial budget and prioritize most of the resources available at our department for several years, in order to compare the performance of different parameterizations of the unsaturated hydraulic conductivity curve (UHCC). This is not realistic.
  This suggestion by the reviewers is better suited to test the performance of all aspects of the numerical model: apart from the Richards's solver, the models for potential evapotranspiration, for reactive solute transport, for plant growth, for root water uptake (including root growth and the response to water and salinity stress), and for nutrient uptake, all come into play. Conducting such a study with one of its objectives being the comparison of UHCC parameterizations, without having these tested previously in the way we did here, does not appear to be a viable pathway.
  There are currently two models that dominate world-wide applied modeling of the real world setting that the reviewer prefers. These are Hydrus-1D and SWAP. Both models are one-dimensional solvers of Richards' equation for unsaturated flow. Both can handle crop growth, albeit with different degrees of sophistication (and input requirements). Both have a track record of multiple decades in modeling unsaturated water flow in scientific and applied settings, as evidenced by the two reviews (Šimůnek et al., 2016; Heinen et al., 2024) we cited in the paper. Both heavily rely on suitable soil water retention curves (SWRCs) and UHCCs of the kind we tested here. In view of the fact that these are the models that are actually used by those applying the science to address real-world problems (SWAP is the model of choice to design irrigation districts and schedule irrigations in real time, for instances), these models have the appropriate complexity, as the reviewer phrases it.

*Unfortunately, I cannot confirm that the presented pure model comparison has much scientific value. Both the choice of the soils and the climate setting appear very much ad-hoc, while other findings appear constraint by using a simple 1d model approach.*

Reply:

The reviewer appears to imply that a multidimensional model would have been better to test the conductivity models. As we explained above, the people that actually use our UHCC parameterizations prefer 1D models. It is those users who we are trying to accommodate with our work. In fact, the teams behind Hydrus-1D and SWAP expressed interest in implementing the junction model UHCC and the associated SWRC, and the writing of the code needed to achieve this is in progress. In our view, this settles the question of the scientific validity of our work.

We are well aware that soils are three-dimensional bodies with considerable spatial variation in its soil hydraulic property curves (SHPs). But assessing the intricacies of this spatial variation is very laborious: a very large number of samples are needed to do so, and the measurements of the SHPs will take many weeks for such a large data set. There are a few multidimensional models available, but we do not see the added value of engaging these models, since none of them allows each individual node to receive its own set of parameter values for the SWRC and UHCC, which would be required to model 3D flow in a heterogeneous soil. At least one of the models allows a simplified representation of soil spatial variability, but using that would negate the reviewers' desire for real-world settings.

We also note that the suggestion for multidimensional flow modeling is at odds with the desire of the reviewer to test the model results against observed data: no methodology exists to measure fluxes within a 3D body of soil. We would have to compare the simulations against fluxes at the top and bottom of a weighing lysimeter, which are by necessity 1D representations of the underlying 3D fluxes. Equifinality in the 3D model of the soil will be inevitable, making the experimental backup of the model and our UHCC questionable.

In combination with the fact that the vast majority of users of the functions we tested will be using them in a 1-D modeling set-up, we do not see the point of testing these expressions on a multi-dimensional model. If a user should wish to use our functions in a multi-dimensional set-up, this will of course be possible. Both to focus a test on an application that is rarely used would reduce its relevance for the scientific community.

The selection of the soils and the climates was deliberate. The various climates ranged from semi-arid to humid. The combination of climates ensured that the numerical model was challenged by having to deal with very large gradients (which occur when rain falls on a very dry soil), very large and very small fluxes, a wide range of soil water contents, reversal of the flow direction, etc.

The soils were chosen based on their very different shapes of the soil water retention curves, and for the different ranges in the unsaturated hydraulic conductivity data, which is an issue in many data sets, reflecting the difficulties of measuring the bulk soil hydraulic conductivity.

The statement that our choices feel ad-hoc (apart from being incorrect) contradicts the preference of the reviewer for tests on real-world data: such a test would make us completely dependent on the availability of weighing lysimeters, since we will not be building one ourselves just to test three UHCCs. We would have no choice at all about soil type, climate, crop rotation, and data acquisition. What is more ad-hoc than that?

*I also regret to tell, that the presentation quality is neither up to the standards required for a top journal nor does it stick to the HESS specific standards. All should parameters named, clearly defined with their units.*

Reply:

We agree that the paper should meet HESS standards. The columns of Tables 1-4 contain all parameters, as well as their units. In response to reviewer 2 we propose to include a Theory section in the revised version (if the editor allows us a revision) that will include the equations of all UHCC parameterizations tested in the paper, as well as the associated parameterization of the SWRC. This will guarantee that all parameters appear in the equations, where they will be explained in full accordance with HESS guidelines.

*The abstract and the introduction are very technical and appear to be written for an "insider".*

Reply:

In the last few years, a lively debate has developed in the literature about SHPs, and theoretical models to parameterize them. Several of these papers appeared in HESS, and we are aware of these and cited them in the preprint. Most of them have the issue that the reviewer mentions, which is natural once a debate conducted in the literature progresses into an advanced state. In these papers, this is solved by referencing the works that allow an interested 'outsider' to follow the discussion. We followed this pattern, established by earlier HESS papers.

Nevertheless, we can review the abstract with this comment in mind, and see if it can be made more accessible, without losing sight of the main contributions of the study. We would prefer to keep the Introduction as it is, because otherwise we would be forced to repeat published literature extensively.

*"A recently introduced junction model with liquid water in either films or capillaries has one parameter less." One parameter less than what?*

Reply:

Less than the additive model introduced in the previous sentence. We will rewrite the sentence to clarify this.

*"We compared calculated water fluxes based on both models and Kosugi's model (which only considers capillary water) by fitting the RIA soil water retention curve and the three conductivity models to data for three soils". What is the RIA soil water retention curve?*

Reply:

That is the RIA parameterization of the SWRC introduced by de Rooij (2022, 2024a), both cited in the preprint. The missing explanation of this acronym is an omission by us that we will rectify if we are allowed to revise the paper.

*The manuscript is also not self-contained. The soil hydraulic model should at least be briefly explained. In this context, I wonder how the authors account of water vapor fluxes. The latter needs proper accounting for the heat balance, as saturated water vapor pressure in soil is a function of temperature. And the water vapor diffusion coefficient is a product of the average free path length and the average thermal velocity of vapor molecules. According to kinetic gas theory, both are functions of temperature as well. Does this imply that the model can only be used in numerical models solving the coupled water and heat balance?*

Reply:

The other reviewer suggested to add the equations, In our reply to that comment we propose to add a Theory section to take care of that. Doing so also address this reviewer's

concerns about the explanation of the various parameters. This will include the water vapor conductivity model, so we can discuss this as well. For details, we refer to the reply to Reviewer 2.

A short answer to answer the reviewer's last question: The temperature effect on water vapor diffusion is accounted for by the model equations. So, yes, the heat flow equation needs to be solved in conjunction with Richards' equation. Both Hydrus-1D and SWAP have this capability. By the way, we have been informed by the author of a multidimensional model that it cannot.

*Using a 1d model to quantify overland flow and removing water after a ponding height of 1cm appear arbitrary, why not 0.5 cm. This approach is simply too simple!*

Reply:

Why is this too simple? It is wonderfully effective in showcasing the effect of the UHCC on the infiltration capacity of the soil. Please bear in mind we are demonstrating, comparing, and testing UHCC models by exposing them to a wide variety of conditions. This is not an application of a numerical model to solve a specific practical problem for one particular stakeholder. And even in applications aimed at solving a real-world problem, this approach is not as uncommon as the reviewer seems to believe.

The statement that we use a 1D model to quantify overland flow is the reviewers', not ours. We chose climates that allowed us to challenge the numerical model by letting it deal with excess water, either because the infiltration rate was exceeded (heavy rains on dry soil), or the storage was exceeded (prolonged heavy rainfall). The paper never states that overland flow estimation was an objective of this study.

*I also wonder about the authors expectation/ hypothesis when comparing the three soils. Which differences would you expect? A comparative plot of the three retention functions would be helpful to formulate a kind of hypothesis.*

Reply:

In a paper on the UHCC, it would be strange to work with only one soil. We selected the soils based on very contrasting shapes of the SWRC: sigmoidal, power law, and a nearly straight wet branch. It is worth noting that the RIA parameterization is the first that is capable of fitting all these shapes, and it is still new. It is therefore difficult to formulate expected behavior a priori. But we were curious, so we set up this simulation study to explore any effects. That is one of the reasons we chose these contrasting soils and exposed them to contrasting climates, with very different infiltration rates and potential evapotranspiration rates. This was indeed intended, and not ad-hoc, as the reviewer claims above.

That being said, the idea of formulating one or several hypotheses is worth considering. Our reply to the next comment indicates how we think this could be implemented.

*I don't get the point of compare different parameter fitting scenarios. You can and should reject fits that drop below a performance certain threshold. Otherwise optional differences in simulated water balance components are contaminated by deficiency of the fits.*

Reply:

We propose to add an explanation of the choice of the set of fixed/fitting parameters. In short:

Set 1 is the most flexible, fitting all parameters. Only the matric potential at oven-dryness is forced to be identical to that of the SWRC, to avoid the presence of liquid water with zero conductivity, or a non-zero conductivity of liquid water while none is present.

Set 2 ensures that the SWRC and the UHCC have similar critical matric potentials: the air-entry value, junction point (where the wet capillary water branch joins the dry film water branch), and the matric potential at oven-dryness are all identical.

Set 3 eliminates one of the terms in the expression for the capillary conductivity, similar to a conductivity model proposed by Assouline (2001), which is cited in the preprint. This will be considerably easier to understand if the equations are included in the paper. This set was adopted because the terms that have $\gamma$ and $\tau$ as exponents have roughly the same effect on the shape of the UHCC. In some cases, this gave fits in which both parameters had extreme values at opposite sides of their limits because one tried to counter the effect of the other, leading the fitting algorithm into an unlikely corner in the parameter space.

Set 4 adopts Mualems's (1976) (cited in the preprint) conductivity model by setting the Kosugi parameters to values defined by him. The remaining parameters (except the matric potential at oven dryness) are fitted.

Set 5 assumes that the parameters fitted for the SWRC have predictive value for the UHCC (the original premise in van Genuchten's (1980) paper, cited in the preprint. Hence, only the parameters that are purely dedicated to the UHCC are fitted.

Both reviewers state that the best fit, determined according to a suitable goodness-of-fit criterion, is by definition the one that ought to be used in simulations. We wanted to know what effect different fits have on calculated fluxes. If the fluxes calculated using different fits are not significantly different, is one fit truly better than the other? What if the best fit involves more fitting parameters than another, but gives (nearly) identical fluxes, would the fit requiring a simpler model or less parameters not be the better choice? It is this kind of rethinking that we think is necessary to determine the optimality of SHP parameterizations. The comments by both reviewers indicate that we need to present this argument more convincingly in the review, should we be allowed to do so. This is best done in the Discussion section, perhaps with a prelude in the revised Materials and Methods where we explain the sets of fitting parameters as proposed above.

In that case, we should also modify the Introduction in order to prepare the reader for the section of the revised Discussion that will be devoted to this topic. The research question (the effect of different fits on calculated fluxes) can perhaps be cast in the form of two alternative hypotheses: 1) the more dissimilar the fits, the more dissimilar the fluxes; and 2) even dissimilar fits can give relatively similar fluxes, owing, for instance to the dampening effect of the vegetation, or any other physical explanation. The results show that there is an effect of the soil on the calculated fluxes, so this also addresses the previous comment of the reviewer. We thank the reviewer for this suggestion, we think it can help us pointing the readership to the key findings.

How exactly we will present all elements of this (hypotheses, revised Methodology, Discussion devoted to the effect of different UHCC parameterizations and different sets of fitting parameters) we do not yet know in detail, because it will require careful writing that we will require a few cycles improvements to finalize.

We note that one of the things we found by not only inspecting the goodness of fit, but also the resulting fluxes, was that the shape of the curve in the dry end hardly affected the

calculated fluxes. The reviewers criticize our approach, but do not object to a result that could only be achieved by this approach.

*The difference between many water balance components are very small, What about defining on simulation/soil type as reference and looking at differences. This will easier reveal differences.*

Reply:

We are not sure that this will work: with different soils across different climates, it will be very difficult to argue convincingly why any of the nine soil-climate combination deserves to be called a reference.

Furthermore if the different UHCC models give very similar water balance components for a particular soil-climate combination, this is a comforting result: the nature of unsaturated flow, and the effect of the vegetation appear to dampen the effect of the soil-vegetation system on the actual fluxes. This would then explain why 1D models historically have been proven to do a good job, despite their simplicity (one of the points the reviewer criticized above). In short, if the simulations give comparable results, this finding alone is sufficient information for someone who is tasked with setting up a model configuration for tackling a practical solution. Zooming in on the smallest differences does not really add much to the analysis.

The long-term water balances tend to be stable: the net infiltration is partitioned over evaporation, transpiration, and deep drainage. For a multi-year water balance, the storage change will typically be close to zero, unless the climate has a clear trend over the simulation period, which was not the case, given the way we set up the weather generator. If the fluxes leaving the soil are initially high, they will drop later, because there is less water available. If they are initially low, they can later be higher. Such self-correcting feedbacks along the temporal axis stabilize the water balance terms. Between climates, there will of course be differences in the portioning of the infiltrating water (Fig. 9, discussed in section 3.4).

Apart from the water balance terms, there are also the time series of the fluxes, shown and discussed in Appendix A. There, difference between the different models are larger, under some conditions. For long-term processes such as groundwater recharge, the long-term water balance is more relevant, but for crop growth, limits to infiltration, etc., smaller time scales become more important. With this in mind, we chose the focus of the discussion of the effect of climate and soils on the various fluxes in Appendix A. The results are plausible: fluxes across the soil surface are dominated by the weather, deep fluxes are more strongly influenced by soil type; in an energy-limited climate (monsoon), there are stronger short-term variations in actual transpiration than in a water-limited climate (semi-arid); across all soils and climates there are clear effects of inter-annual climatic variations on the various fluxes.

Since we modeled hypothetical scenarios, and not a practical problem, we believe there is little point in delving deep into the intricacies of the boundary conditions. Instead, we prefer that the focus remains on the SHP-driven response to these conditions.

---

## Author Comment (AC2)

Comments of reviewer 2

Comments in *italics*, reply in regular font.

**General comments**

*The manuscript provides a functional evaluation of three unsaturated hydraulic conductivity models, namely the Kosugi (KGV), additive (ADV), and junction model (JUV), in simulating the water balance under different soils and climate conditions. To this end, multi-year simulations are conducted with the HYDRUS 1D model, and the results are compared in terms of aggregated mean fluxes at both the upper and bottom boundaries of the soil profile. The selected functional approach is noteworthy as it allows to compare soil hydraulic models with different characteristics (i.e., considering vapor diffusion) directly in terms of their effects on simulated outputs. However, it should be pointed out that despite being fairly conducted, the study is basically a numerical exercise with a limited link with real data either in terms of the selected soils or simulated water fluxes. In fact, the soil hydraulic conductivity data are gathered from literature while the simulated fluxes are not compared with measured ones, and, therefore, their validity can only be assessed in relative terms. Thus, this study adds a limited contribution beyond what was already investigated in previous studies conducted by the same author(s).*

Reply:
      The main role of the soil water retention curve (SWRC) and the unsaturated hydraulic conductivity curve (UHCC) is to define the soil hydraulic properties (SHPs) for numerical solvers of Richards' equation. A test of different parameterizations by examining the fluxes calculated by such a solver is intrinsically relevant for this reason alone. Our reply to Reviewer 1 (who made a similar suggestion) explains the vast cost and effort required for a test involving measured data, as well a some of its pitfalls.
      In the literature, the vast majority of comparisons of different parameterizations of SHPs limit themselves to evaluating the fits to data points of the SWRC and/or the UHCC, and completely ignore the effect of the different fits on the fluxes calculated by the Richards' solvers for which they are chiefly developed. Evidence of this can be found in the papers reviewed by Assouline and Or (2013, cited in the preprint) for the UHCC, and by Khlosi et al. (2008) and Madi et al. (2018, cited in the preprint) for the SWRC.
      For the modeling community that runs Richards' solvers for practical applications, it is important to know what kind of parameterizations are available, and how they perform. For modelers, performance in terms of the goodness of fit against measured conductivity is less important if the resulting fluxes calculated by different fits are not that different. It is highly relevant for this community if a parameterization with a comparable or even somewhat better fit than its competitors leads to more frequent crashes of their numerical model. The reviewer did not consider this aspect, which naturally requires test runs with a numerical model. We selected the scenarios in such a way that the numerical model needed to negotiate a wide range of forcings, and was confronted with the numerically notoriously difficult case of infiltration into dry soil. The scenarios chosen were, therefore, not at all ad hoc, as reviewer 1 suspects.
      To our knowledge, comparisons of parameterizations of the SHPs by testing them using numerical models for unsaturated flow is exceedingly rare. Apart from this paper, we found only two: Ippisch et al. (2006), who used as a test case a single 2D case study of a two-layer soil with macropores, and an earlier paper by the second author with coworkers (de Rooij et al., 2021, cited in the preprint). This study uses a modification of the set-up of de Rooij et al. (2021) that is more thorough, encompassing 135 test cases in total. The reviewer

does not acknowledge the dearth of numerical tests of SHP parameterizations like the one we performed, and does not seem to appreciate the very large number of test cases covered in this paper.

 ***Specific comments***
*The presentation of the selected hydraulic conductivity models is poor and their properties are difficult to understand if the readers do not refer to the previous studies. I think the manuscript's readability could be improved if the models were presented, even showing the related equations. There are several symbols in text and tables (e.g., $h_j$, $K_{sc}$, $K_{sa}$, $\tau$, $\gamma$) that are not defined when they are introduced for the first time or not defined at all. Furthermore, the choice of using the Kosugi model for comparison should be motivated as well as the advantage of ADV and JUV over classical models should be highlighted.*

Reply:
       We found guidance in rules 5 and 6 of the Obligations for authors on the HESS website, specifically the following elements:
- An author should cite those publications that have been influential in determining the nature of the reported work and that will quickly guide the reader to the initial work essential for understanding the present investigation.
- Fragmentation of research papers should be avoided. A scientist who has done extensive work on a system or group of related systems should organize publication so that each paper gives a complete account of a particular aspect of the general study.

       The referencing is complete, save for the explanation of the acronym RIA for the SWRC parameterization we used, and the references linking to the KRIAfitter code. These oversights by us, we will of course correct in case we are allowed to revise the paper. With those corrections, the information necessary to follow the paper can be readily found from open access sources. That being said, we believe the suggestion to include the underlying equations is good, and will increase the readability of the paper without undue repetition of already published material. If a review is granted, we will propose to add a Theory section after the Introduction, where equations for the RIA parameterization for the SWRC and the three UHCC parameterizations will be provided and briefly explained, backed up by adequate referencing to the two source papers (de Rooij, 2022, 2024b, cited in the preprint). This will also resolve the unclarities of the parameters, as the reader will then be able to see where they appear in the equations.
       The selection of the three UHCC parameterizations was based on the findings of de Rooij (2024b). An explanation will be provided.  In short, the explanation will be along the following line:
The Kosugi model (KGV) is a generalization of Mualem's model, which has been the most popular model for the past 4 decades, supplemented with a model for the equivalent conductivity that represents the diffusion of water vapor. This model only considers capillary water and water vapor. The additive model (ADV) represents a generalization of the PDI model (introduced in Peters and Durner (2008), Peters (2013), and Weber et al. (2019) – all cited in the preprint) that accounts for water adsorbed in films as well. De Rooij (2024b) elucidated the implicit underlying assumptions (and a flaw) of PDI. He developed alternative models by introducing other, equally (or more) plausible assumptions, and found that the flaw was impossible to fully eliminate. He also found evidence of overparameterization, which is one of the reasons he introduced the simpler junction model (JUV), which has the same number of fitting parameters as KGV, and one less than ADV, but still accounts for water in films. The alternatives to ADV developed by de Rooij (2024b) did not perform better, as demonstrated by de Rooij (2025) (cited in the preprint), so there is little incentive to further

pursue them. Hence, the model of choice for the past 40+ years (KGV), its challenger with a more comprehensive description of the UHCC (ADV), and the more parsimonious alternative to ADV (JUV) are the models chosen for the test of their performance when used in a numerical models of soil water flow.

*I hardly understand the rationale for considering the different combinations of fixed and fitted parameters in model simulations. Indeed, if the soil hydraulic conductivity data are measured, the best choice is the one that yields the lowest values of fitting statistics (RMSE, R2, AIC, MBE....), and comparing alternative strategies in terms of model outputs is trivial without independent reference data. However, it should be recognized that the selection of the parameter fitting strategy is a crucial step once the experimental data are obtained and guidelines helping practitioners in this choice are probably lacking but, in my opinion, this point should be addressed with larger soil databases including soil with different origins and characteristics.*

Reply:
      Reviewer 1 also wondered about the different sets of fitting parameters. We refer to out reply there for that part of this comment. We also explain there our reasoning for examining the effects of these fits on the calculated fluxes in addition to comparing goodness-of-fit measures.
      As explained above, the reviewer underestimates the time, effort, and resources needed to acquire independent reference data. Furthermore, a model-based comparison revealing effects of parameterizations and the choice of fitting parameters has intrinsic value: it shows if and how the choice of UHCC and its set of fitting parameters affects calculated fluxes. There are two possible outcomes: significant effect and hardly any effect. Both are tangible, actionable results. Especially in the case of the former, independent data would be very welcome, but in case of extreme differences, expert judgment can be used to assess the plausibility of the calculated fluxes. In case of the latter, the simplest model is probably the best.
      This paper is about a numerical test of three UHCC models. For the extensive test that the reviewer requests, involving fits of UHCC parameterizations on a wide range of soils, we refer to the 780 best fits in total presented by de Rooij (2025), cited in the preprint.

**Minor comments**

*L.11: What is RIA?*

Reply:
      The other reviewer also noted this omission. As explained in our response there, will rectify this if we are allowed to revise the paper.

*L.49-64: Is the discussion conducted here strictly necessary for the study? I checked that these points were already discussed in the papers where the models were developed.*

Reply:
      This section summarizes the more elaborate discussions to explain the backgrounds of the tested UHCC parameterizations, and their relation to one another. We can shorten it of course, or refer the reader to these papers and leave it out altogether. However, elsewhere, the reviewer asks for more information about the UHCC parameterizations and asks us to include far more information from these papers. If we do that (as we intend to do), it seems a bit awkward to leave out this short segment of text.

L.71: On parameter less than what?

Reply:
Well, one parameter less than the more complicated models that were discussed in the text preceding this sentence (and which the reviewer wishes us to remove). We will clarify this if we are allowed to revise the paper. The proposed modification reads: '…than the more complicated models derived by de Rooij (2024a) based on those by Durner and Peters (2008), Peters (2013), and Weber et al. (2019).'

L.84: "all of them including diffusive movement of water vapor": is it true also for KSG model? If yes, how the water diffusion was accounted for in KSG model?

Reply:
We do not use the acronym KSG in the paper. Our model KGV adds the vapor conductivity of Peters (2013) as modified by de Rooij (2024a) to Kosugi's model for the conductivity of capillary water, as explained in de Rooij (2024a). In the revised version with the added Theory section, this will be clear to see directly from the equations.

L.153: What is the KRIAfitter code?

Reply:
Thank you for noticing this. We omitted two references there: de Rooij (2024c, 2025). Both were already cited, and discussed in some detail, in the Introduction, but the link with the KRIAfitter code was not made there. The references will be added.

L.155-163: Most of this information was already given in L.90-103. I suggest to unify these sections.

Reply:
Good suggestion, thank you. We think it is best to keep the reference to the earlier paper in L. 90-92, and transfer the rest of that section to L. 155-163, removing any overlap in the process.

L.214-215: Is this conclusion of general validity? I don't think so given only three soils were considered.

Reply:
Three very different soils, three very different climates, and a total of 135 model runs totaling 1350 simulated years (excluding 810 years of burn-in). This is by far the most extensive numerical test of UHCC parameterizations to date.

**Reference**
Khlosi, M., Cornelis, W. M., Douaik, A., van Genuchten, M. Th. and Gabriels, D.: Performance evaluation of models that describe the soil water retention curve between saturation and oven dryness, Vadose Zone J., 7, 87-96. doi:10.2136/vzj2007.0099, 2008.